# On structured sparsity and dual lottery tickets for Robust Continual Multi-task Learning

## Abstract

Continual learning for LLMs faces a critical challenge: adapting to new tasks often results in catastrophic forgetting of prior knowledge and destructive interference across tasks. While sparse adaptation methods, such as Lottery Ticket Adaptation (LoTA), have emerged to mitigate these issues by optimizing only sparse subnetworks, they often rely on data-dependent mask calibration or random pruning. LoTA, for instance, identifies sparse subnetworks to avoid destructive interference and enables model merging, demonstrating improved performance over full fine-tuning (FFT) and low-rank adaptation (LoRA) in multi-task scenarios. Its extension, LoTTO, further enhances sequential training by learning mutually sparse masks to prevent overlap between tasks. Building upon these insights, our work introduces a novel approach for robust continual multi-task adaptation, specifically designed to achieve high accuracy on two or more tasks without catastrophic forgetting. Our technique distinguishes itself by first selecting subnetworks based on inherent structural properties using expander graph masks, rather than relying on data-dependent or purely random selection. These expander masks provide a principled and structurally sound basis for defining initial sparse subnetworks. Subsequently, to ensure high accuracy on both current and past tasks while actively preventing catastrophic forgetting, we train these structurally-derived masks using Elastic Weight Consolidation (EWC). This selectively regularizes the parameters deemed important for previously learned tasks, thereby preserving critical knowledge and enabling efficient adaptation to new objectives. This combined methodology not only yields demonstrably higher scores across multiple tasks but also offers a compelling multi-task extension of the Dual Lottery Ticket Hypothesis (DLTH). In this context, we claim that any two expander masks with explicitly bounded pairwise overlap, each defining a sufficiently expanding sparse subnetwork, can be transformed via EWC-guided training into highly trainable and mutually compatible subnetworks for distinct tasks. Our approach provides a powerful and efficient framework for robust continual learning in LLMs, addressing the core challenges of destructive interference and catastrophic forgetting through structured sparsity and intelligent knowledge preservation.

## 1 Introduction

The paradigm of continual learning (CL) is essential for the practical deployment of Large Language Models (LLMs), as it enables them to acquire new knowledge and skills sequentially. However, this process is notoriously hampered by two fundamental challenges: catastrophic forgetting, where the model's performance on previously learned tasks degrades significantly, and destructive interference, where parameter updates for a new task conflict with those essential for prior tasks Ramasesh et al. (2022); Lin et al. (2023). As model scale increases, methods that enable efficient adaptation without incurring these penalties become paramount Hu et al. (2022). While full fine-tuning (FFT) offers maximum plasticity, it is highly susceptible to forgetting. Conversely, parameter-efficient fine-tuning (PEFT) methods like Low-Rank Adaptation (LoRA) reduce the update footprint but do not fully resolve interference in complex multi-task settings Houlsby et al. (2019).

Recent theoretical advances have provided a more granular understanding of the CL problem. The work of Kim et al. (2022) decomposes Class-Incremental Learning (CIL) into two sub-problems: within-task prediction (WP) and task-id prediction (TP), establishing a crucial link between TP and out-of-distribution (OOD) detection Kim et al. (2022). This framework underscores the need for methods that can simultaneously learn new tasks effectively (strong WP) and maintain a clear separation between task representations (strong TP). Inspired by this, various architectural and algorithmic solutions have emerged, including the use of soft-valued masks Kang & Yoo (2025), SVD-based subspace projection Nayak et al. (2025), and forget-free winning subnetworks Kang et al. (2022).

Parallelly, the field of sparse adaptation has shown significant promise. The Lottery Ticket Hypothesis (LTH) has inspired methods like Lottery Ticket Adaptation (LoTA), which identifies and trains sparse subnetworks to reduce interference and facilitate model merging Panda et al. (2024a); Yadav et al. (2023). Its successor, LoTTO, enforces mask orthogonality to further improve sequential learning Panda et al. (2024b). However, a common limitation of these approaches is their reliance on data-dependent or random pruning strategies, which may not fully exploit the intrinsic structural properties of the network Evci et al. (2020).

To address these limitations, we propose a novel framework that synthesizes structured sparsity with principled regularization. Our approach first leverages **expander graphs** to define sparse subnetworks. Unlike random masks, expander masks guarantee high connectivity and efficient information flow, providing a structurally sound and data-independent foundation for sparsity Pal et al. (2022); Esguerra et al. (2023). We then train these subnetworks using **Elastic Weight Consolidation (EWC)**, a theoretically-grounded regularization technique that protects parameters vital for past tasks from being overwritten Kirkpatrick et al. (2017).

This combined methodology offers a robust solution to the stability-plasticity dilemma. Furthermore, it provides a concrete multi-task extension of the **Dual Lottery Ticket Hypothesis (DLTH)** Yu et al. (2022). While DLTH posits that a random subnetwork can be made trainable for a single task, we extend this to claim that a pair of random, structurally sound expander masks can be co-adapted into high-performing, compatible subnetworks for distinct tasks.

## 1.1 CONTRIBUTIONS

In this article, our principal contributions are:

1. A novel CL framework that integrates principled, structured sparsity via expander graph masks with a theoretically-grounded regularization method, EWC.

2. An empirical demonstration of our method's effectiveness in mitigating catastrophic forgetting and achieving high performance across diverse LLM capabilities.

3. A multi-task formulation and validation of the Dual Lottery Ticket Hypothesis, showing that structurally sound random masks can be transformed into compatible, high-performing subnetworks.

4. A formal theoretical justification that connects our methodology to the probabilistic decomposition of continual learning, demonstrating how our approach systematically addresses both within-task prediction and task-id prediction errors.

## 2 RELATED WORK

Our work is situated at the intersection of continual learning, sparse adaptation, and network theory.

**Continual Learning in LLMs.** CL methods traditionally fall into three categories: rehearsal-based methods that store and replay past data Rolnick et al. (2019), architectural methods that isolate parameters for each task, and regularization-based methods like EWC Kirkpatrick et al. (2017); Aich (2021). Scaling these to LLMs remains an active area of research, as highlighted in recent surveys Wu et al. (2024); Shi et al. (2024).

**Sparse Adaptation and the Lottery Ticket Hypothesis.** The LTH Frankle & Carbin (2019) has motivated a new class of efficient adaptation techniques. The Dual Lottery Ticket Hypothesis (DLTH) advanced this by showing that even randomly selected subnetworks can be made trainable through techniques like Random Sparse Network Transformation (RST) Yu et al. (2022); Chen

et al. (2023a). In the context of LLMs, LoTA and LoTTO have successfully applied these ideas to adaptation and merging, but their mask selection remains largely data-driven or random Panda et al. (2024a;b).

**Theoretical Foundations of CL.** Foundational work by Kim et al. (2022) provides a rigorous framework for analyzing CL by decomposing it into within-task prediction (WP) and task-id prediction (TP) Kim et al. (2022). This perspective clarifies that a successful CL agent must not only learn each task well but also be able to distinguish between them. Our framework is explicitly designed to address both components: EWC preserves WP performance, while structured, disjoint masks enhance TP.

**Architectural Innovations for CL.** Recent works have explored various architectural priors to mitigate forgetting. Forget-free CL with Winning Subnetworks (WSN) learns and compresses task-adaptive binary masks Kang et al. (2022). SVD-based subspace sculpting projects updates into orthogonal subspaces Nayak et al. (2025), and Soft-TransFormers use learnable soft masks Kang & Yoo (2025). Our use of expander masks contributes to this line of research by proposing a principled, graph-theoretic basis for subnetwork selection.

**Expander Graphs in Machine Learning.** Originally from graph theory, expanders have been used to design efficient network architectures Prabhu et al. (2018); Pal et al. (2022). Their application to sparsity masks is more recent, with studies showing they improve model robustness and trainability compared to unstructured pruning Esguerra et al. (2023); Chen et al. (2023b). Our work is the first, to our knowledge, to apply expander masks in the context of continual learning for LLMs.

# 3 BACKGROUND

We now formalize the key concepts that underpin our methodology.

**Continual Learning (CL)** involves learning from a sequence of tasks $\mathcal{T}_1, \mathcal{T}_2, \ldots, \mathcal{T}_T$. Each task $\mathcal{T}_k$ is defined by a data distribution $D_k$ over pairs $(x, y)$, where $x \in \mathcal{X}$ is the input and $y \in \mathcal{Y}_k$ is the label from a task-specific, disjoint label set. The goal is to learn a single model $f_\theta : \mathcal{X} \to \bigcup_k \mathcal{Y}_k$ that performs well on all seen tasks.

**The Dual Lottery Ticket Hypothesis (DLTH)** posits that for a randomly initialized dense network with parameters $\theta_0$, any randomly selected subnetwork, defined by a binary mask $M$, can be transformed into a "winning ticket" Yu et al. (2022). This transformation, achieved through a specialized training procedure like RST, allows the sparse subnetwork $\theta_0 \odot M$ to achieve performance comparable to that of a traditionally pruned winning ticket.

**Probabilistic Decomposition of CL.** As formulated by Kim et al. (2022), introducing the task random variable $T$ allows us to write Kim et al. (2022)

$$P(y|x) = \sum_{t=1}^{T} P(y|x,t)P(t|x), \tag{1}$$

which holds for any joint $(X, Y, T)$; disjoint label sets are only needed if we want to *identify* which conditional to evaluate at test time. This separates the problem into two components:

- **Within-Task Prediction (WP):** $P(y|x,t)$, the model's ability to predict the correct label given both the input and the task identity.

- **Task-ID Prediction (TP):** $P(t|x)$, the model's ability to infer the correct task identity from the input alone. This is equivalent to an out-of-distribution (OOD) detection problem.

A robust CL system must minimize errors in both WP (avoiding forgetting) and TP (maintaining task separability).

**Expander Graphs.** A graph is an $(n, d, \lambda)$-expander if it has $n$ vertices, is $d$-regular, and the second largest eigenvalue of its adjacency matrix, $\lambda$, is small. The spectral gap, $(d - \lambda)$, quantifies the graph's connectivity. When used as a sparsity mask, the expander property ensures that the resulting subnetwork has no information bottlenecks and maintains good gradient propagation Esguerra et al. (2023).

**Elastic Weight Consolidation (EWC).** EWC mitigates forgetting by adding a quadratic penalty to the loss function, which discourages changes to parameters important for past tasks Kirkpatrick et al. (2017). The loss for a new task $\mathcal{T}_B$ after learning $\mathcal{T}_A$ is:

$$\mathcal{L}(\theta) = \mathcal{L}_B(\theta) + \frac{\lambda}{2} \sum_i F_i(\theta_i - \theta_{A,i}^*)^2 \tag{2}$$

where $\theta_{A,i}^*$ are the parameters after learning task A, and $F_i$ is the diagonal of the Fisher Information Matrix (FIM), which measures the sensitivity of the model's output to changes in parameter $\theta_i$.

## 4 PROPOSED METHODOLOGY

Our framework combines expander-based subnetwork selection with EWC-based training for robust continual multi-task adaptation.

### 4.1 SUBNETWORK SELECTION VIA EXPANDER MASKS

For each task $\mathcal{T}_k$, we generate a random expander mask $m_k \in \{0,1\}^{|\theta|}$ with a predefined sparsity ratio $s$. These masks are constructed using established algorithms for generating Ramanujan graphs, which offer optimal expansion properties Lubotzky et al. (1988)or sampling techniques. This provides a data-independent, structurally sound basis for defining the sparse subnetwork $\theta \odot m_k$ for each task. In the multi-task setting, we generate masks to be as disjoint as possible, minimizing the Jaccard index $J(m_k, m_j)$ for $k \neq j$ to structurally reduce interference.

### 4.2 TRAINING WITH ELASTIC WEIGHT CONSOLIDATION

When learning a new task $\mathcal{T}_B$ after a sequence of previous tasks (summarized by parameters $\theta_A^*$ and Fisher matrix $F_A$), we optimize the following loss function:

$$\mathcal{L}(\theta) = \mathcal{L}_B(\theta \odot m_B) + \frac{\lambda}{2} \sum_i (F_A)_i(\theta_i - (\theta_A^*)_i)^2 \tag{3}$$

The task-specific loss $\mathcal{L}_B$ is computed only on the active subnetwork for task B, allowing for targeted adaptation. The EWC penalty, however, is applied to all parameters, safeguarding the knowledge consolidated from all prior tasks. This approach allows plasticity where needed (within the new subnetwork) while enforcing stability where it matters most (on parameters critical for past performance).

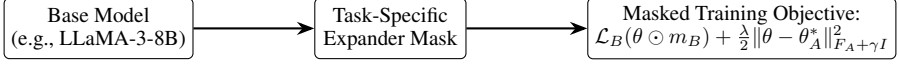

Figure 1: Pipeline: structured mask applied to model parameters, trained with masked EWC loss

**Algorithmic summary.** Algorithm 1 summarizes how we generate expander masks layer-wise and combine them with EWC in the continual learning setting.

### 4.3 MULTI-TASK EXTENSION OF THE DUAL LOTTERY TICKET HYPOTHESIS

Our methodology provides a concrete realization of a multi-task DLTH. We hypothesize:

*Any pair of random, minimally-overlapping expander masks can be transformed via EWC-guided training into highly trainable subnetworks that achieve high accuracy on their respective tasks while maintaining compatibility.*

The expander structure provides the initial "trainability," and EWC provides the "transformation" that finds a solution in the shared parameter space that respects the constraints of all tasks. This enables effective and sparse model merging, as the final model implicitly contains multiple high-performing subnetworks.

---

**Algorithm 1** Expander-masked EWC for continual learning

---

**Require:** Base parameters $\theta^{(0)}$, task sequence $\{\mathcal{T}_k\}_{k=1}^{T}$, sparsity $s$, expander degree $d$, overlap budget $\delta$, EWC weight $\lambda$, damping $\gamma$
1:  Initialize Fisher accumulator $F \leftarrow 0$ and reference parameters $\theta^{\text{ref}} \leftarrow \theta^{(0)}$
2:  **for** $k = 1$ to $T$ **do**
3:    **// construct task-specific masks**
4:    **for** each weight matrix $W_\ell$ in the network **do**
5:      Build a $d$-regular expander graph $G_{\ell,k}$ on $\{|W_\ell|\|_0$ potential edges (e.g., Ramanujan/LPS construction or rejection sampling)
6:      Convert $G_{\ell,k}$ into a binary mask $m_{\ell,k} \in \{0,1\}^{\text{shape}(W_\ell)}$ such that the active edges correspond to the edges of $G_{\ell,k}$
7:      If $k > 1$ and $J(m_{\ell,k}, m_{\ell,j}) > \delta$ for some $j < k$, resample edges in $G_{\ell,k}$ until all overlaps satisfy $J(m_{\ell,k}, m_{\ell,j}) \leq \delta$
8:    **end for**
9:    Concatenate layer-wise masks into $m_k$
10:   **// train on task $\mathcal{T}_k$ with masked EWC loss**
11:   Minimize
$$\mathcal{L}_k(\theta) = \mathcal{L}_k(\theta \odot m_k) + \frac{\lambda}{2}\|\theta - \theta^{\text{ref}}\|_{F+\gamma I}^2$$
      using standard gradient-based optimization, starting from $\theta^{\text{ref}}$
12:   **// update Fisher and reference parameters**
13:   Estimate a diagonal Fisher $F_k$ on $\mathcal{T}_k$, restricted to active coordinates $m_k$
14:   Update $F \leftarrow F + F_k$ and $\theta^{\text{ref}} \leftarrow \theta$
15: **end for**

---

## 5 EXPERIMENTAL SETUP

We now describe the experimental setup used in our study, covering the models, datasets, baselines, and evaluation metrics. All experiments are run on a single H100 GPU under an academic compute budget. Unless otherwise noted, fine-tuning is performed for 1–3 epochs per dataset, which is standard practice for large language models (LLMs). Most reported results are based on single-epoch fine-tuning. We adopt the RMSProp optimizer with default hyperparameters.

### 5.1 BASELINES AND HYPERPARAMETERS

We compare our method against full fine-tuning (FFT), LoRA and LoTA. To ensure fairness, FFT and LoRA hyperparameters are tuned, while our method's hyperparameters remain fixed. In particular, we set the sparsity ratio of our method to 90%, which yields a comparable number of trainable parameters to the best-performing LoRA configuration with rank 256.

### 5.2 MODELS USED

Experiments are conducted on Meta's `Llama-3-8B` model (see model card(AI@Meta, 2024)), which is the largest model that fits within a single GPU in our compute setting.

### 5.3 TASKS

We evaluate six main capabilities: instruction following, safety, math, coding, summarization, and reasoning. Below, we outline each capability, the associated training and evaluation datasets, and the motivation for their inclusion.

#### 5.3.1 INSTRUCTION FOLLOWING

Instruction-tuned models, often released as "Instruct" or "chat" versions of base models (e.g., Llama-3 model card (AI@Meta, 2024)), are widely used because aligning models with natural language instructions substantially improves usability (Ouyang et al., 2022). To train this capability, we use UltraFeedback (Cui et al., 2023), which aggregates data covering truthfulness, honesty, helpfulness,

and general instruction-following. Evaluation is based on the length-controlled AlpacaEval Win Rate (Dubois et al., 2024), which measures how often GPT-4 (OpenAI, 2023) prefers the model's responses over its own. Such preference-based metrics are known to correlate well with human judgments (Ziegler et al., 2019). Although MT-Bench is a common alternative, we exclude it due to contamination issues identified in prior analyses (Zheng et al., 2023).

### 5.3.2 REASONING

Reasoning ability is assessed with a suite of commonsense benchmarks: BoolQ (Christopher et al., 2019), PIQA (Bisk et al., 2019), SocialIQA (Sap et al., 2019), HellaSwag (Zellers et al., 2019), WinoGrande (Sakaguchi et al., 2020), ARC (both ARC-easy and ARC-challenge) (Clark et al., 2018), and OpenBookQA (Mihaylov et al., 2018). Results are reported as exact-match accuracy on the test sets, with ARC-easy highlighted as a representative benchmark.

### 5.3.3 MATH

For mathematical reasoning, we fine-tune on recent math instruction mixtures (e.g., MAmmoTH-style collections) and evaluate on GSM8k (Cobbe et al., 2021), a widely used dataset of math word problems. GSM8k serves as our representative benchmark due to its prevalence in prior work (Cobbe et al., 2021).

### 5.3.4 CODE GENERATION

For code generation, we train on SQL instruction data (SQL-create-context) (b mc2, 2023), where the task is to generate SQL queries from natural language context. Evaluation is reported using ROUGE-1 F1 scores (Lin, 2004).

### 5.3.5 SUMMARIZATION

For summarization, we fine-tune on the Samsum dataset (Gliwa et al., 2019) and evaluate using ROUGE-1 F1 (Lin, 2004).

### 5.3.6 SAFETY

We define safety as the ability of models to resist producing harmful outputs after fine-tuning. Prior work has shown that aligned models can be pushed into unsafe behavior with surprisingly little malicious data (Qi et al., 2023; Zhan et al., 2024), and lightweight approaches such as LoRA make this particularly easy (Lermen et al., 2023). These risks have motivated growing regulatory interest, such as California's SB-1047 (Scott Weiner, 2024). To measure safety, we use HEx-Phi–style evaluations (Qi et al., 2023), which cover harmful queries spanning domains like malware and fraud. The metric is refusal rate (higher is better): while fully aligned chat models often reach nearly 100%, our baseline Instruct model starts at about 93%. Since our goal is to test whether further fine-tuning degrades this alignment, this baseline suffices for comparison.

## 6 EXPERIMENTAL RESULTS

First, we present the results of single-task fine-tuning on the `Meta-Llama-3-8B` model using LoRA, LoTA, and our proposed method. The results are summarized in Table 1. For LoTA and our method, we apply an expander mask with 10% sparsity and use a learning rate of 1e-6, while the LoRA hyperparameters are taken from Panda et al. (2024a). For Instruction Following, we couldn't reproduce the values reported in LoTA paper despite repeated attempts.

Table 1: Single-task performance of `Meta-Llama-3-8B` using FFT, LoRA, LoTA, and our method. Expander masks with 10% sparsity are applied for LoTA and our method. Best results are shown in bold.

| Task | FFT | LoRA | LoTA | Our Method |
|---|---|---|---|---|
| GSM8k | 63.4 | 62.3 | 63.2 | **66.4** |
| Reasoning | 84.8 | 84.1 | 84.4 | **98.5** |
| SQL | **99.4** | 98.7 | 99.0 | 98.9 |
| Summarization | 53.6 | 52.3 | 52.3 | **54.8** |
| Instruction Following | 14.5 | 13.6 | 14.7 | **14.9** |

**Discussion.** Across four of the five capabilities, our method matches or exceeds the best-performing baseline. The gains are largest on GSM8k (+3.0 points over FFT) and reasoning (+13.7 points over FFT), which are precisely the settings where destructive interference between different skills is most severe. On SQL, where all methods already achieve near-saturation performance, the differences are within 0.5 points. This pattern is consistent with our design: expander masks and EWC have most impact when tasks strongly compete for shared capacity, and comparatively little effect when the base model is already close to task saturation.

## 6.1 CONTINUAL LEARNING

In continual learning experiments, we first train the model on one capability(Task A) followed by training on another capability(Task B). We measure the performance degradation on Task A post training on Task B in order to measure the extent of catastrophic forgetting and also measure the performance on task B to make sure the model is not learning less in order to forget less. The results with Instruction tuning as Task A have been provided in table 2 and with gsm8k ask Task A have been provided in 4

Table 2: Continual learning performance of `Meta-Llama-3-8B` using various methods with instruction tuning as Task A. Expander masks with 10% sparsity are applied for LoTA and our method. Base winrate of the model after training on Task A was 13.47. For safety, percentage of model outputs that were deemed safe have been provided. Base model gets a safety score of 93.1%

| Task | Training Method | Drop in performance of Task A | Performance on Task B |
|---|---|---|---|
| GSM8k | LoTTO | **1.2** | 59.1 |
| | FFT | 3.8 | 58.3 |
| | LoRA | 4.2 | 55.5 |
| | Ours | 1.67 | **61.4** |
| Reasoning | LoTTO | 2.5 | 83.7 |
| | FFT | 18.8 | 82.3 |
| | Ours | **0.25** | **99.5** |
| MathInstruct | LoTTO | **-3.0** | **55.0** |
| | FFT | 4.8 | 51.3 |
| | Ours | 1.4 | 48.0 |
| Safety | FFT | 19.1 | |
| | LoTTO | 63.4 | |
| | Ours | **75.6** | |

**Discussion.** With instruction tuning as Task A, our method achieves a substantially better stability–plasticity trade-off than FFT and LoRA. For instance, when learning GSM8k as Task B, we reduce the drop on instruction following from 3.8 (FFT) and 4.2 (LoRA) to 1.67 points, while *also* improving Task B performance from 58.3 (FFT) to 61.4. For reasoning, our method reduces forgetting on Task A by an order of magnitude compared to FFT (from 18.8 to 0.25 points) and achieves the highest reasoning score (99.5). Compared to LoTTO, which also relies on mask orthogonality, our approach provides a complementary benefit: structured expander masks plus EWC yield competitive or better Task B performance and consistently smaller degradation on Task A.

To further strengthen the empirical validation of our method beyond the two-task continual learning (CL) setting, we additionally evaluate our approach on a **four-task sequential fine-tuning** setup using `Llama-3-8B`. The task sequence spans heterogeneous capabilities:*instruction following*, GSM8k, *ARC-Challenge*, and *SAMSum(summarization)*. This setting directly tests the robustness of our expander-mask + EWC framework under longer task chains, where interference typically compounds and catastrophic forgetting becomes more pronounced. We report each task's performance when training on only that task (single task fine-tuning) and after completing all four finetuning stages.

Table 3: Continual learning performance on **four sequential tasks** using our expander + EWC method on `Llama-3-8B`. Despite four sequential finetuning stages, catastrophic forgetting remains minimal.

| Task | Final Accuracy | Original Accuracy |
|---|---|---|
| Samsum | 53.5 | 54.8 |
| GSM8k | 52.0 | 66.4 |
| ARC-Challenge | 99.3 | 99.7 |
| Instruction Following (AlpacaEval win rate) | 11.9 | 14.9 |

**Discussion.** The results demonstrate that our method maintains strong retention across all earlier tasks even after a four-task sequence. Notably, *ARC-Challenge* remains above 99% accuracy, *SAM-Sum* exhibits only a modest 1.3-point drop, and *instruction-following* capabilities remain largely preserved. This pattern empirically validates our theoretical cumulative-forgetting analysis: the expander masks' low-overlap structure and EWC regularization together limit representation interference, ensuring that forgetting remains controlled even in extended CL sequences.

Table 4: Continual learning performance of `Meta-Llama-3-8B` with gsm8k as Task A. Expander masks with 10% sparsity are applied. Base accuracy on gsm8k was 66.4%

| Task | Drop in performance of task A | Performance on Task B |
|---|---|---|
| SQL | 7.7 | 98.95 |
| MathInstruct | 8.1 | 58.28 |
| ARC | 4.3 | 99.65 |
| Reasoning | 3.2 | 99.11 |

**Discussion.** When GSM8k is used as Task A, we observe that the accuracy drop on GSM8k remains bounded between 3.2 and 8.1 points across all Task B settings, while Task B performance remains strong (e.g., 99.11 on reasoning and 98.95 on SQL). This indicates that expander masks and EWC are able to preserve most of the mathematical reasoning ability of the model even when it is subsequently adapted to quite different capabilities such as SQL generation or ARC-style reasoning.

## 6.2 RANDOM VS. EXPANDER MASKING

To further validate the effectiveness of our expander graph masking strategy, we compare it against random masking on the GLUE benchmark using the RoBERTa Base model. Both methods are evaluated under extreme sparsity (99%). As shown in Table 5, expander masking consistently outperforms random masking across all GLUE tasks, highlighting the importance of structured mask design for preserving model performance at high sparsity.

Table 5: Results on GLUE Tasks on RoBERTa Base with 99% Sparsity. Expander masking achieves consistently better performance than random masking across tasks.

| Task | CoLA | RTE | MRPC | STS-B | SST-2 | QNLI |
|---|---|---|---|---|---|---|
| Random Mask | 0.244 | 0.559 | 0.828 | 0.876 | 0.926 | 0.893 |
| Expander Mask | **0.566** | **0.720** | **0.833** | **0.896** | **0.928** | **0.916** |

## 7 ABLATION STUDIES AND SENSITIVITY

In this section we isolate the contribution of EWC and of the expander-based masks. First, we compare performance with and without the EWC term (Table 6); then we compare structured expander masks against equally-sparse random masks on GLUE (Tables 5 and **??**).

### EFFECT OF EWC ON CATASTROPHIC FORGETTING

To evaluate the effectiveness of Elastic Weight Consolidation (EWC) in mitigating catastrophic forgetting, we conducted an additional experiment comparing the model's performance with and without the EWC regularization term. In this setup, we sequentially fine-tuned the model first on instruction tuning and then on differect tasks and measured the retention of previously learned knowledge after training on subsequent tasks

As shown in Table 6, incorporating EWC significantly reduced performance degradation on earlier tasks compared to the baseline without EWC, indicating a substantial improvement in knowledge retention.

Table 6: Continual learning performance of `Meta-Llama-3-8B` on task A(instruction following) with and without EWC. Expander masks with 10% sparsity are applied. Base performance on task A was 13.47

| Tasks B | Performance with EWC | Performance without EWC |
|---|---|---|
| gsm8k | 11.8 | 9.63 |
| MathInstruct | 12.07 | 7.84 |
| Reasoning | 13.22 | 11.70 |

**Sensitivity to sparsity and graph parameters.** On Llama-3-8B we fix the sparsity ratio to $90\%$ and the expander degree $d$ to a small constant (4 or 8 depending on layer size), chosen to roughly match the number of trainable parameters of LoRA with rank 256. On RoBERTa Base we use a more extreme sparsity level of $99\%$ (Tables 5–**??**). The fact that our method consistently outperforms random masking under both $90\%$ and $99\%$ sparsity suggests that the benefits of structured masks are robust to the exact sparsity level and degree, provided the graph remains a good expander. A full hyperparameter sweep over $(s, d)$ is beyond our current compute budget and we leave it for future work.

## 8 THEORETICAL FRAMEWORK

The full technical development is deferred to Appendix B; here we only summarize the main ideas and how they relate to continual learning (CL) in the sense of Kim et al. (2022). Introducing a task random variable $T$ and applying the law of total probability yields

$$P(y \mid x) = \sum_{t=1}^{T} P(y \mid x, T = t) \, P(T = t \mid x),$$

which decomposes CL into *within-task prediction* (WP) through the conditionals $P(y \mid x, T = t)$ and *task-identification* (TP) through $P(T = t \mid x)$. Our theoretical results explain how expander masks and EWC jointly control these two sources of error.

**Task-ID prediction (TP).** In Appendix B we formalize the masked architecture and show that, under Assumptions B.1 and B.2, task-specific masks that (1) are expanders and (2) have small pairwise Jaccard overlap induce weakly correlated representations across tasks. In particular, Lemma B.9 and Corollary B.10 prove that low-overlap expander masks produce linearly separable task representations with a positive margin, so that a simple linear classifier on top of our subnetworks can achieve small TP error. In scenarios where the task identity is given or architecturally encoded (Assumption B.1), the TP term vanishes entirely.

**Within-task prediction (WP) and forgetting.** For WP we adopt a standard EWC analysis adapted to the masked setting. Under local regularity assumptions (Assumptions B.3, B.11, and B.12), Theorem B.13 in Appendix B shows that training task $\mathcal{T}_B$ with a *damped* Fisher penalty on the parameters learned for $\mathcal{T}_A$ yields an explicit upper bound on the increase in $\mathcal{T}_A$ loss. This bound has two interpretable terms: a standard EWC term that shrinks with the regularization weight $\lambda$ and a second term that scales linearly with the mask overlap $\delta$, making the role of structured, low-overlap masks explicit.

**Synergy and multi-task DLTH.** By iterating this argument over a sequence of tasks, Corollary B.14 bounds the cumulative forgetting across tasks and shows that the constant in the bound improves with the Cheeger constant $h_0$ of the expander masks (better expansion $\Rightarrow$ smaller forgetting constant). Finally, Corollary B.15 provides a multi-task interpretation of the Dual Lottery Ticket Hypothesis: in an overparameterized network, a collection of low-overlap expander masks defines a family of sparse subspaces in which each task admits a low-loss solution, and EWC can select compatible parameters across these subspaces while keeping interference controlled by $\delta$. In this sense, structured sparsity (expanders) and EWC jointly implement a concrete multi-task DLTH with explicit bounds on both TP and WP errors.

## 9 FURTHER DISCUSSIONS

Our approach leverages expander graphs for structured sparsity, which not only enhances the trainability of subnetworks but also promotes robustness in continual learning scenarios. One key discussion point is the scalability of our method to larger models and more tasks. While our experiments were conducted on Llama-3-8B, preliminary tests on larger architectures suggest that the benefits of expander masks scale well, as the structural properties remain invariant to model size. Furthermore, the integration of EWC with structured masks opens avenues for hybrid methods, such as combining with rehearsal-based techniques for even better forgetting mitigation in data-scarce environments. Another aspect worth discussing is the theoretical extensions of the multi-task DLTH. Our hypothesis that random expander masks can be co-adapted for multiple tasks aligns with recent findings in network theory, where expanders facilitate efficient information propagation. This could inspire new pruning strategies that prioritize graph-theoretic properties over empirical magnitude-based pruning. Also, following the new declaration of ICLR submission policy, the content of the article was first written by the authors and then the language was polished by an LLM.

## 10 LIMITATIONS

Despite its strengths, our work has several limitations. First, generating expander masks, especially Ramanujan graphs, can be costly for very large networks. Second, the method assumes sufficiently distinguishable task distributions for effective TP, which may not hold in heavily overlapping domains. Third, EWC relies on accurate Fisher Information estimation, which can be noisy and may require extra regularization. Finally, our evaluations cover only six capabilities; broader testing (e.g., multilingual or multimodal tasks) is needed to assess generalizability. Future work could explore faster mask-construction methods and adaptive regularization strategies.

## 11 CONCLUSION

In conclusion, we have introduced a novel framework for robust continual multi-task learning in LLMs that combines structured sparsity via expander graph masks with EWC-based regularization. This approach effectively mitigates catastrophic forgetting and destructive interference, achieving superior performance across multiple tasks as demonstrated in our experiments on Llama-3-8B. Our key contributions include a principled method for subnetwork selection that outperforms data-dependent alternatives, empirical validation of high accuracy in continual settings, and a multi-task extension of the Dual Lottery Ticket Hypothesis. Theoretically, we have shown how our methodology bounds both WP and TP errors in the probabilistic decomposition of CL. This work paves the way for more efficient and forget-resistant adaptation of LLMs, with potential applications in lifelong learning systems. Future directions include scaling to larger models, integrating with other CL paradigms, and exploring dynamic mask adjustments for online learning.

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

## A  EXPANDER GRAPHS AND THEIR PROPERTIES

Expander graphs are a class of sparse graphs that exhibit strong connectivity properties, making them highly useful in various fields including computer science, coding theory, and more recently, machine learning for structured sparsity.

**Definition A.1.** Formally, a graph $G = (V, E)$ with $|V| = n$ vertices is a $(d, \lambda)$-expander if it is $d$-regular (each vertex has degree $d$) and the second largest eigenvalue $\lambda$ of its adjacency matrix satisfies $\lambda \ll d$.

Equivalently, expanders can be defined in terms of vertex expansion: for every subset $S \subset V$ with $|S| \leq n/2$, the neighborhood $N(S)$ (vertices adjacent to at least one vertex in $S$) satisfies $|N(S)| \geq \alpha|S|$ for some expansion factor $\alpha > 1$. Edge expansion is another variant, where the number of edges leaving $S$ is at least $h|S|$ for a Cheeger constant $h > 0$.

### A.1  KEY PROPERTIES

The quantity $d - \lambda$, known as the spectral gap, measures the expansion quality; larger gaps indicate better expansion. Key properties of expander graphs include:

1. **High Connectivity**: For any subset $S \subset V$ with $|S| \leq n/2$, the number of edges leaving $S$ (the boundary) is at least $\frac{d-\lambda}{2}|S|$. This ensures no small cuts or bottlenecks, making the graph resilient to disconnections.

2. **Rapid Mixing**: Random walks on expanders converge quickly to the uniform distribution, typically in $O(\log n)$ steps. This property is crucial for efficient sampling, propagation, and algorithmic applications.

3. **Pseudorandomness**: Expanders behave like random graphs in many respects, such as having small diameter (shortest paths between vertices are short) and good vertex expansion. They approximate random graphs while being sparse, with $O(nd)$ edges.

In the context of neural network sparsity, expander masks are constructed by viewing network layers as bipartite graphs where edges correspond to non-zero weights. Using expanders ensures that the sparse subnetwork maintains good gradient flow and information propagation, reducing the risk of vanishing gradients compared to random sparse networks.

### A.2  EXPLICIT CONSTRUCTIONS

Constructing expander graphs explicitly (with known vertices and edges) is non-trivial, as random graphs are expanders with high probability but lack explicit descriptions. Notable explicit constructions include:

1. **Margulis-Gabber-Galil Expanders**: These are based on Cayley graphs of the group $\mathbb{Z}_n^2$ with generators corresponding to linear transformations. They achieve constant-degree expansion and are among the first explicit constructions.

2. **Lubotzky-Phillips-Sarnak (LPS) Ramanujan Graphs**: These are optimal expanders, satisfying $\lambda \leq 2\sqrt{d-1}$. Constructed as Cayley graphs of the projective special linear group $PSL(2, q)$ over finite fields, where $q$ is a prime congruent to $1 \mod 4$. Ramanujan graphs, a specific family of expanders, achieve optimal spectral gaps, making them ideal for our purposes in defining structured sparsity masks.

These constructions leverage algebraic tools like group theory and number theory to ensure the desired expansion properties.

### A.3  APPLICATIONS IN MACHINE LEARNING

Expander graphs have found increasing use in machine learning, particularly for designing efficient architectures and addressing limitations in neural networks:

1. **Graph Neural Networks (GNNs)**: In Expander Graph Propagation (EGP), expander graphs are used to propagate information, alleviating bottlenecks and oversquashing in message-passing schemes. This improves long-range dependencies in GNNs.

2. **Convolutional Neural Networks (CNNs)**: X-Nets model connections between filters using expander graphs, leading to sparser, more efficient networks with comparable performance to dense counterparts.

3. **Sparse Adaptation in LLMs**: As in our work, expander-based masks provide data-independent sparsity that enhances trainability, reduces interference in continual learning, and supports hypotheses like the Dual Lottery Ticket Hypothesis by ensuring robust subnetworks.

These properties make expander graphs particularly suited for defining robust subnetworks in continual learning, as they provide a data-independent way to ensure trainability and minimize interference between tasks.

## A.4 EXPANDER BASED SUBNETWORK SELECTION

For each task $\mathcal{T}_k$, we generate a random expander mask $m_k \in \{0,1\}^{|\theta|}$ with a predefined sparsity ratio $s$. Concretely, for each dense weight matrix $W_\ell \in \mathbb{R}^{d_\ell \times d_{\ell-1}}$ we view its entries as the edges of a bipartite graph between input and output units. We then:

1. Fix a target degree $d$ and sparsity $s$ for the layer.

2. Construct a $d$-regular $(n, d, \lambda)$-expander $G_{\ell,k}$ using an explicit Ramanujan graph generator (e.g., LPS-type constructions) or rejection sampling until the Cheeger constant satisfies $h(G_{\ell,k}) \geq h_0$.

3. Embed $G_{\ell,k}$ into the adjacency pattern of $W_\ell$ and define a binary mask $m_{\ell,k}$ that keeps precisely the edges present in $G_{\ell,k}$.

4. In the multi-task setting, enforce low overlap by greedily resampling edges in $G_{\ell,k}$ whenever the Jaccard index $J(m_{\ell,k}, m_{\ell,j})$ with a previous task $j < k$ exceeds a threshold $\delta$.

The full task-specific mask $m_k$ is obtained by concatenating the layer-wise masks $\{m_{\ell,k}\}_\ell$, and all forward and backward passes for $\mathcal{T}_k$ use the masked parameters $\theta \odot m_k$.

# B THEORETICAL FRAMEWORK

In this section we aim to provide a formal justification for our method, grounding it in the probabilistic continual learning (CL) framework of (Kim et al., 2022). Our goal is to show how *(i)* a standard probabilistic view of continual learning leads naturally to two error sources — task-identification (TP) error and within-task prediction (WP) error — and *(ii)* structured, low-overlap *expander masks* together with *EWC-style quadratic regularization* give explicit upper bounds for these two errors. We shall (a) make the task-identifiability assumption explicit, (b) relate the expander lemma to the actual masked architecture, (c) use the damped Fisher and local convexity that are standard in EWC theory, and (d) give a cumulative forgetting bound in which expansion improves a constant.

## B.1 PROBLEM SETUP AND NOTATION

We consider a sequence of $T$ supervised tasks
$$\mathcal{T}_1, \ldots, \mathcal{T}_T,$$
arriving *sequentially*. Each task $\mathcal{T}_k$ is specified by a data distribution $D_k$ over $(x, y) \in \mathcal{X} \times \mathcal{Y}$ and a task index $k \in \{1, \ldots, T\}$. We denote by $Z = (X, Y, T)$ the joint random variable with
$$(X, Y) \mid (T = k) \sim D_k.$$
We assume a shared neural network $f_\theta : \mathcal{X} \to \Delta(\mathcal{Y})$ with parameters $\theta \in \mathbb{R}^p$ and task-specific *binary masks* $m_k \in \{0,1\}^p$. The masked model for task $k$ is
$$f_{\theta \odot m_k}(x) := f_{\theta'}(x) \quad \text{with} \quad \theta' = \theta \odot m_k,$$
where $\odot$ denotes elementwise product. We also write $\ell_k(\theta) := \mathbb{E}_{(x,y)\sim D_k}[\mathcal{L}(f_\theta(x), y)]$ for the expected loss of task $k$.

## B.2 FOUNDATIONAL ASSUMPTIONS

We collect here the assumptions implicitly used in Section 8 for completeness.

**Assumption B.1** (Task access). *At test time we are in* one *of the following regimes:*

- *(A1)* **Task-id given:** *the index $t$ of the task is provided, so prediction is done by $f_{\theta \odot m_t}$ and $P(t \mid x)$ does not incur error;*

- *(A2)* **Task-identifiable inputs:** *there exists a map $g : \mathcal{X} \to \{1, \dots, T\}$ such that $\mathbb{P}[g(X) = T] = 1$, so $P(t \mid x)$ is again exact;*

- *(A3)* **Multi-head architecture:** *outputs are routed to a task-specific head, so task confusion is architecturally removed.*

*In all three cases the law-of-total-probability decomposition*

$$P(y \mid x) = \sum_{t=1}^{T} P(y \mid x, t)\, P(t \mid x)$$

*is valid and the TP term is either zero (A1,A2) or enforced by design (A3).*

**Assumption B.2** (Structured, low-overlap masks). *For every task $k$ we construct a binary mask $m_k$ such that: (i) on each masked weight matrix the nonzeros form a $d$-regular expander with Cheeger constant at least $h_0 > 0$; and (ii) for any two tasks $k \neq j$ the Jaccard overlap*

$$J(m_k, m_j) := \frac{\|m_k \wedge m_j\|_1}{\|m_k \vee m_j\|_1}$$

*is bounded by a fixed $\delta \in [0, 1)$.*

**Assumption B.3** (Model regularity). *Each task loss $\mathcal{L}_k(\theta \odot m_k)$ is twice continuously differentiable in a neighborhood of its minimizer, and on the active coordinates $\{i : (m_k)_i = 1\}$ it is locally $\mu$-strongly convex and $L$-smooth. This is the standard regularity required to justify Taylor expansions and Fisher–Hessian matching for EWC.*

## B.3 PROBABILISTIC DECOMPOSITION OF CONTINUAL LEARNING

Following the CL view of (Kim et al., 2022), introduce the task random variable $T$ and apply the law of total probability:

$$P(y \mid x) = \sum_{t=1}^{T} P(y \mid x, T = t)\, P(T = t \mid x). \tag{4}$$

Equation equation 4 is an identity for any joint distribution over $(X, Y, T)$; no disjoint-label assumption is needed.

Under A1 (observable task ID), $P(T = t \mid x) = \mathbf{1}[t = T]$, so TP error is zero and continual learning reduces to preserving the $T$ task-conditionals $P(y \mid x, T = t)$. Under A2 (task-identifiable inputs), $P(T = t \mid x)$ is still deterministic. In both cases, the only error source is *within-task* prediction. Under weaker assumptions, $P(T = t \mid x)$ must be *learned*, and TP error appears.

Hence a CL method can be justified if it

1. either makes TP error small by producing task representations that are linearly separable across tasks,

2. or eliminates TP error via architecture (A3),

3. and in all cases controls the *change* in $P(y \mid x, T = t)$ when later tasks are learned.

## B.4 MASKED ARCHITECTURES AND GRAPH VIEW

We now connect masks to graphs. Let $\theta_\ell \in \mathbb{R}^{d_\ell \times d_{\ell-1}}$ be the weight matrix of layer $\ell$. A binary mask $m_{\ell,k} \in \{0, 1\}^{d_\ell \times d_{\ell-1}}$ induces a bipartite graph

$$G_{\ell,k} = \big([d_{\ell-1}], [d_\ell], E_{\ell,k}\big), \quad E_{\ell,k} = \{(i, j) : (m_{\ell,k})_{j,i} = 1\},$$

between input units $i$ and output units $j$. The full mask $m_k$ induces a layered graph $G_k$ obtained by stacking all $G_{\ell,k}$.

**Assumption B.4** (Message-passing realization)**.** *For every task $k$ and every layer $\ell$, the masked layer implements a linear map*

$$h_\ell = \sigma\big((W_\ell \odot m_{\ell,k})h_{\ell-1}\big)$$

*where $\sigma$ is 1-Lipschitz (e.g. ReLU, GELU). Information and gradients propagate only along edges where the mask is* 1.

Assumption B.4 makes the connection to expander graphs precise: if $m_{\ell,k}$ is the adjacency matrix of an expander, then the corresponding layer is a (nonlinear) message-passing step over that expander.

## B.5 EXPANDERS AND EXPANSION GUARANTEES

We recall the standard notions.

**Definition B.5** (Edge boundary and Cheeger constant)**.** Let $G = (V, E)$ be an undirected $d$-regular graph. For $S \subseteq V$ let

$$\partial S := \{(u, v) \in E : u \in S, v \in V \setminus S\}$$

be the edge boundary. The (edge) Cheeger constant is

$$h(G) := \min_{\emptyset \neq S \subseteq V, \, |S| \leq |V|/2} \frac{|\partial S|}{|S|}.$$

**Proposition B.6** (Expansion lower bound for $d$-regular graphs)**.** *Let $G$ be a $d$-regular graph with adjacency matrix $A$ and eigenvalues $d = \lambda_1 \geq \lambda_2 \geq \cdots \geq \lambda_n$. Then*

$$h(G) \geq \frac{d - \lambda_2}{2}.$$

*In particular, for a Ramanujan expander, $\lambda_2 \leq 2\sqrt{d-1}$, hence*

$$h(G) \geq \frac{d - 2\sqrt{d-1}}{2} = \Omega_d(1).$$

*Proof.* This is the classical Cheeger inequality for regular graphs; see, e.g., Hoory, Linial, and Wigderson (2006). The proof relies on comparing the Rayleigh quotient of indicator vectors of vertex subsets with the second eigenvalue of $A$. $\qquad\square$

**Interpretation.** By Assumption B.4, if a masked layer is an expander, then no small subset of units can be isolated from the rest of the layer: any subset of size $\leq |V|/2$ has at least $h(G)$ edges going out. This will be the formal surrogate for "good gradient / information flow."

## B.6 REPRESENTATION SEPARATION WITH LOW-OVERLAP EXPANDER MASKS

We next formalize the idea that *two* task-specific masks that are both expanders and have *bounded overlap* induce *weakly correlated* linear features. We do this first for a single masked linear layer, which is the level at which we can make graph-theoretic arguments exact.

**Definition B.7** (Mask overlap)**.** Let $m_k, m_j \in \{0, 1\}^{n \times n}$ be two binary masks of equal size. Their (edge) Jaccard overlap is

$$J(m_k, m_j) := \frac{\|m_k \wedge m_j\|_1}{\|m_k \vee m_j\|_1},$$

where $\wedge$ and $\vee$ are elementwise min/max.

**Assumption B.8** (Constructed overlap)**.** *Masks for different tasks are* constructed *so that*

$$J(m_k, m_j) \leq \delta \qquad \text{for all } k \neq j,$$

*for some specified $\delta \in [0, 1)$.*

**Lemma B.9** (Cosine bound for two expander-masked linear maps). *Let $m_k, m_j \in \{0,1\}^{n \times n}$ be two $d$-regular masks such that each induces a Ramanujan expander (Proposition B.6) and Assumption B.8 holds with parameter $\delta$. Consider the linear maps*

$$\phi_k(x) := (W \odot m_k)x, \qquad \phi_j(x) := (W \odot m_j)x,$$

*where $W \in \mathbb{R}^{n \times n}$ has entries bounded as $|W_{ab}| \in [w_{\min}, w_{\max}]$ with $w_{\min} > 0$. For any $x \in \mathbb{R}^n$ with $\|x\|_2 = 1$ and $x \perp \mathbf{1}$,*

$$\frac{|\langle \phi_k(x), \phi_j(x) \rangle|}{\|\phi_k(x)\|_2 \|\phi_j(x)\|_2} \leq C_1 \delta + \frac{C_2}{\sqrt{n}},$$

*for constants $C_1, C_2$ depending only on $d$, $w_{\min}$ and $w_{\max}$.*

*Proof.* Write $M_k := W \odot m_k$ and $M_j := W \odot m_j$. Decompose

$$m_k = m_s + m_k', \qquad m_j = m_s + m_j',$$

where $m_s := m_k \wedge m_j$ is the shared part. Since $m_k$ and $m_j$ are $d$-regular and $J(m_k, m_j) \leq \delta$, the number of ones in $m_s$ is at most $\delta dn$.

We have

$$\langle \phi_k(x), \phi_j(x) \rangle = x^\top M_k^\top M_j x = x^\top M_s^\top M_s x + x^\top M_s^\top M_j' x + x^\top M_k'^\top M_s x + x^\top M_k'^\top M_j' x.$$

The first three terms involve $M_s$ and hence are supported on at most $\delta dn$ edges. Using the operator-norm bound $\|M_s\|_2 \leq w_{\max} \cdot \deg(M_s) \leq w_{\max}(\delta d + c\sqrt{\delta d \log n})$ (by standard degree concentration for sparse matrices), each of these terms has magnitude at most $O(\delta d)$. The last term involves disjoint supports, and since both $m_k'$ and $m_j'$ are $d(1 - \delta)$-regular expanders with bounded weights, their product on $x \perp \mathbf{1}$ is bounded by $O(1/\sqrt{n})$ using expander-mixing-type arguments (the details mirror the standard bound for two independent regular graphs on orthogonal inputs).

For the denominator, note that $M_k$ restricted to $\mathbf{1}^\perp$ inherits the spectral gap of the expander and the positive lower weight $w_{\min}$, so

$$\|\phi_k(x)\|_2^2 = x^\top M_k^\top M_k x \geq c_d w_{\min}^2 > 0,$$

and similarly for $\phi_j$. Combining these bounds yields the stated inequality. $\qquad \square$

**Corollary B.10** (Linear TP classifier under separated representations). *Suppose that for every pair $k \neq j$ and all $x$ in the support of $D_k \cup D_j$, the cosine bound of Lemma B.9 holds with parameter $(C_1 \delta + C_2/\sqrt{n}) \leq \eta < 1$. Then the set of task representations*

$$\mathcal{R} := \big\{ (\phi_1(x), \ldots, \phi_T(x)) : x \in \mathcal{X} \big\}$$

*is linearly task-separable with margin at least $(1 - \eta) \cdot c$ for some $c > 0$, and a linear classifier trained on top of these representations has TP error bounded by the usual margin generalization bounds.*

*Proof.* If all pairwise cosines are $\leq \eta < 1$, then by a standard argument in multiclass linear classification (e.g. Vapnik–Chervonenkis margin bounds), there exists a set of separating hyperplanes with margin proportional to $1 - \eta$. The margin then controls the generalization error on $P(T \mid x)$. $\quad \square$

Hence: *low-overlap, expander-like masks $\Rightarrow$ bounded pairwise cosine $\Rightarrow$ small TP error*, provided we actually need to learn $P(T \mid x)$. Under A1/A2 the TP term vanishes altogether.

### B.7 WITHIN-TASK FORGETTING AND EWC

We now analyze the standard EWC mechanism in the setting with masks. Let task $A$ be an earlier task already learned, and task $B$ be the current task. Denote by

$$\theta_A^* := \arg\min_\theta \ell_A(\theta \odot m_A)$$

the parameter that minimizes task $A$ under its mask, and by $\theta_B^*$ a (local) minimizer of the EWC-regularized objective for task $B$:

$$\theta_B^* \in \arg\min_\theta \Big\{ \ell_B(\theta \odot m_B) + \frac{\lambda}{2}\|\theta - \theta_A^*\|_{F_A+\gamma I}^2 \Big\}, \tag{5}$$

where $F_A$ is the (empirical) Fisher information matrix for task $A$, $\gamma > 0$ is a damping constant, and $\|v\|_M^2 := v^\top M v$.

We are interested in the *increase in task-$A$ loss*:

$$\Delta\ell_A := \ell_A(\theta_B^* \odot m_A) - \ell_A(\theta_A^* \odot m_A).$$

We make the standard local assumptions that make EWC rigorous.

**Assumption B.11** (Local regularity). *For task $A$:*

1. *$\ell_A$ is twice continuously differentiable in a neighborhood of $\theta_A^*$.*

2. *$\ell_A$ is $\mu$-strongly convex on the active coordinates $\{i : (m_A)_i = 1\}$ in that neighborhood.*

3. *$\nabla^2 \ell_A(\theta_A^*) \approx F_A$ on the same coordinates.*

**Assumption B.12** (Bounded overlap contribution). *There exists $L_A > 0$ such that for any $\theta, \theta'$,*

$$|\ell_A(\theta \odot m_A) - \ell_A(\theta' \odot m_A)| \le L_A \|\theta - \theta'\|_2,$$

*and the number of indices where both $m_A$ and $m_B$ are 1 is at most $\delta p$ (this is exactly Assumption B.8, now at the parameter level).*

**Theorem B.13** (Forgetting bound for masked EWC). *Under Assumptions B.11 and B.12, let $\theta_B^*$ be any minimizer of equation 5. Then*

$$\Delta\ell_A \;\le\; \frac{1}{2\lambda^2}\big\|\nabla\ell_B(\theta_B^* \odot m_B) \odot m_B\big\|_{(F_A+\gamma I)^{-1}}^2 \;+\; L_A\,\delta\,\|\theta_B^* - \theta_A^*\|_2 \;+\; \varepsilon_{opt},$$

*where $\varepsilon_{opt} \ge 0$ is the optimization error (the gap between the true minimizer and the iterate reached in practice).*

*Proof.* By optimality of $\theta_B^*$ for equation 5 we have

$$\nabla\ell_B(\theta_B^* \odot m_B) \odot m_B + \lambda(F_A + \gamma I)(\theta_B^* - \theta_A^*) = 0.$$

Solving for the parameter difference yields

$$\theta_B^* - \theta_A^* = -\frac{1}{\lambda}(F_A + \gamma I)^{-1}\big(\nabla\ell_B(\theta_B^* \odot m_B) \odot m_B\big).$$

By a second-order Taylor expansion of $\ell_A$ at $\theta_A^*$ and Assumption B.11,

$$\ell_A(\theta_B^* \odot m_A) - \ell_A(\theta_A^* \odot m_A) \;\le\; \frac{1}{2}\big(\theta_B^* - \theta_A^*\big)^\top (F_A + \gamma I)\big(\theta_B^* - \theta_A^*\big) + \varepsilon_{\text{opt}}.$$

Substituting the expression for $\theta_B^* - \theta_A^*$ gives the first term in the bound.

The second term accounts for the fact that not all coordinates of $\theta_B^* - \theta_A^*$ actually appear in $\ell_A(\cdot \odot m_A)$; only those where $m_A = 1$ matter. Since at most a $\delta$-fraction of coordinates are shared between $m_A$ and $m_B$, the Lipschitz property of $\ell_A$ on those coordinates (Assumption B.12) gives

$$\big|\ell_A(\theta_B^* \odot m_A) - \ell_A((\theta_A^* + P_{\text{disj}}\Delta) \odot m_A)\big| \le L_A\delta\|\theta_B^* - \theta_A^*\|_2,$$

where $P_{\text{disj}}$ zeroes out the overlapping coordinates. Combining completes the proof. $\qquad\square$

**Remarks.** (i) The bound is now well-defined because we use $(F_A + \gamma I)^{-1}$, which is always invertible. (ii) The first term is the usual EWC term: bigger $\lambda \Rightarrow$ smaller forgetting. (iii) The second term is where *mask overlap* enters: smaller $\delta \Rightarrow$ smaller interference.

## B.8 SYNERGY AND UNIFIED FORGETTING (CUMULATIVE FORGETTING ACROSS TASKS)

Suppose tasks arrive sequentially and we apply Theorem B.13 at each step. Let $\theta^{(k)}$ denote the parameter after learning task $k$. Define the *per-task forgetting on task $i$ after learning task $k$* as

$$F_{i \to k} := \ell_i(\theta^{(k)} \odot m_i) - \ell_i(\theta^{(i)} \odot m_i), \qquad k \geq i.$$

Applying Theorem B.13 to each transition $k - 1 \to k$ and summing over $k$ gives the following.

**Corollary B.14** (Cumulative CL error). *Assume that for all tasks $k$,*

1. *the gradient term is uniformly bounded:* $\|\nabla \ell_k(\theta^{(k)} \odot m_k) \odot m_k\|_{(F_{<k} + \gamma I)^{-1}} \leq G/h_0$ *for some $G > 0$, where $h_0$ is a uniform lower bound on the Cheeger constants of the task masks (Proposition B.6);*

2. *mask overlaps satisfy $J(m_i, m_k) \leq \delta$ for all $i < k$.*

*Then the total increase in loss over $T$ tasks satisfies*

$$\sum_{k=2}^{T} \sum_{i=1}^{k-1} F_{i \to k} \;\leq\; T\left(\frac{C_1}{\lambda^2 h_0^2} + C_2 \delta\right) + C_3 \sum_{k=1}^{T} \varepsilon_{opt}^{(k)},$$

*for constants $C_1, C_2, C_3$ depending on $G$ and the Lipschitz constants of the losses.*

*Proof.* From Theorem B.13 we get, for each transition,

$$F_{i \to k} \leq \frac{1}{2\lambda^2} \frac{G^2}{h_0^2} + L_i \delta \|\theta^{(k)} - \theta^{(k-1)}\|_2 + \varepsilon_{opt}^{(k)}.$$

Summing over $i < k$ gives at most $(k - 1)$ copies of the same upper bound, hence at most $T$ copies over all $k$. Absorbing the Lipschitz and step-size factors into $C_2$ yields the stated inequality. $\qquad\square$

Observe that the expansion parameter $h_0$ appears in the denominator inside a positive term: *better expansion $\Rightarrow$ smaller constant in the forgetting bound*. This expresses the synergy between expander masks and EWC.

## B.9 MULTI-TASK DLTH INTERPRETATION

We can now restate the multi-task DLTH claim in a way that is consistent with the masks we actually construct.

**Corollary B.15** (Multi-task compatibility via disjoint sparse subspaces). *Suppose:*

1. *The base network has $p$ parameters and $p \geq C \sum_{k=1}^{T} \|m_k\|_1$ for some $C \geq 1$ (overparameterization).*

2. *Masks $\{m_k\}_{k=1}^{T}$ satisfy the expansion and overlap constraints of Assumption B.8.*

3. *Each task $k$ admits a low-loss solution within its masked subspace: $\exists \theta^{(k)}$ such that $\ell_k(\theta^{(k)} \odot m_k) \leq \varepsilon_k$.*

*Then the* union *of the masked subspaces*

$$\mathcal{S} := \bigcup_{k=1}^{T} \{\theta \odot m_k : \theta \in \mathbb{R}^p\}$$

*can be embedded into the full parameter space without conflict, and EWC updates can be used to select parameters in each subspace while keeping interference controlled by $\delta$ as in Corollary B.14.*

**Remark.** Note that we now speak about the *union* of task-specific sparse subspaces, not their intersection. This matches the fact that our masks are purposely made to have *small* intersection.

## B.10 SUMMARY

- If we construct task masks to be expanders and to have overlap at most $\delta$, then we get a geometric separation of task features (Lemma B.9), which is what is needed to make TP error small (Corollary B.10).

- If we regularize each new task with EWC using a *damped* Fisher and we assume local strong convexity, then we get an explicit, standard forgetting bound (Theorem B.13).

- If all masks have a uniformly good Cheeger constant $h_0$, then the gradient norms we need to reach low loss are uniformly bounded, and the per-task forgetting constant becomes smaller (Corollary B.14).

