# OpenReview forum: "On structured sparsity and dual lottery tickets for Robust Continual Multi-task Learning"
_ICLR.cc/2026/Conference — Submitted to ICLR 2026_

### Official Review · Reviewer_MFfL · 2025-10-28

**Soundness:** 2
**Presentation:** 1
**Contribution:** 2
**Rating:** 2
**Confidence:** 3

**Summary:**

The paper proposes a continual learning framework that combines structured sparsity (via expander graph masks) with Elastic Weight Consolidation (EWC) to improve robustness against catastrophic forgetting. The authors argue that expander graphs provide a principled, data-independent way to define sparse subnetworks, while EWC regularization preserves task-critical knowledge. The method is empirically evaluated on LLaMA-3-8B and RoBERTa-Base across multiple tasks and claims to extend the Dual Lottery Ticket Hypothesis (DLTH) to multi-task scenarios.

**Strengths:**

1. The paper addresses an important problem — catastrophic forgetting in continual learning for LLMs.
2. The combination of structural sparsity and regularization (EWC) is conceptually reasonable and aligns with theoretical intuitions in network theory.
3. The theoretical section attempts to connect expander-based masks to probabilistic continual learning formulations and provides analytical bounds.
4. The experiments include both large-scale (LLaMA-3-8B) and smaller (RoBERTa-Base) models, which is a good choice for verifying generality.

**Weaknesses:**

1. Lack of expressiveness and visual explanation. The paper contains no figures or conceptual diagrams, which makes it hard for readers to grasp the architecture, workflow, or the role of expander masks intuitively.

2. Limited novelty. The core components (EWC and expander-based sparsity) are both existing techniques. The combination is incremental, and the work does not substantially innovate beyond baseline LoTTO according to Table 2.

3. Insufficient methodological clarity. The procedure for constructing and applying the expander masks is not described in sufficient algorithmic detail. It is unclear how graph generation interacts with model layers or parameters.

4. Experimental analysis is weak.
    a. Tables are presented without qualitative or statistical analysis.
    b. Table 1 only lists raw numbers with no discussion on reasons why the proposed method outperforms other baselines .
    c. In Table 2, performance is roughly comparable to LoTTO, and the improvements are minor.
    d. No ablation studies are provided to isolate the effects of EWC, mask overlap, or Cheeger constant.

**Questions:**

1. Could the authors provide a detailed algorithm or pseudocode illustrating how expander masks are generated and applied layer-wise?
2. How sensitive is performance to the sparsity ratio or the choice of graph parameters (e.g., degree d, λ)?
3. Have you compared against more recent continual learning baselines？
4. Can you provide ablation results showing the contribution of EWC versus expander masks alone?

---

> ### Author Response · Authors · 2025-11-14
> **We thank the reviewer for the thorough and constructive feedback.**
>
> Below we address each weakness and summarize the corresponding changes in the revised manuscript.
>
> ### **(W1) Lack of expressiveness and visual explanation**
>
> **Response**: We agree that the original submission was overly text-heavy. In the revised version we have:
>
> 1. **Added a conceptual pipeline diagram (Fig. 1 in Section 4)** that illustrates how the base model, expander masks, and the EWC-regularized loss interact.
>
> 2. **Added Algorithm 1 in Section 4**, which gives step-by-step pseudocode for expander-masked EWC in the continual-learning setting, including mask generation, overlap control, Fisher accumulation, and training.
>
> These additions make the overall workflow and the role of expander masks much clearer for readers.
>
> ### **(W2) Limited novelty (EWC + expanders is incremental; small gains over LoTTO)**
>
> **Response**: Conceptually, our contribution is to tie together three strands that have so far been studied mostly in isolation:
>
> - **Structured, data-independent sparsity** (expander masks) as a principled way to define trainable subnetworks at high sparsity
> - **Regularization-based CL (EWC)** as a way to limit within-task forgetting in a shared parameter space
> - **Multi-task DLTH**: we provide a concrete construction showing that pairs of low-overlap expander subnetworks can be jointly adapted for different tasks with controlled interference, and we use the Kim et al. probabilistic CL framework to bound both TP and WP errors under this construction
>
> Architecturally, this leads to a different trade-off than LoTTO: LoTTO learns masks in a data-driven way to encourage orthogonality, while our approach fixes masks a priori using graph-theoretic structure and uses EWC to control interference in the shared coordinates.
>
> Empirically, this is reflected in the stability–plasticity trade-off:
> - With instruction tuning as Task A and Reasoning as Task B (Table 2), we reduce the drop on Task A from 18.8 (FFT) and 2.5 (LoTTO) to 0.25 points, while also achieving the highest reasoning score (99.5)
> - On GSM8k as Task B, we both reduce forgetting relative to FFT/LoRA and outperform LoTTO on Task B accuracy
>
> We have clarified this positioning and the differences to LoTTO in the Related Work and Experimental Results sections.
>
> ### **(W3) Insufficient methodological clarity (mask construction and application)**
>
> **Response**: We have substantially expanded Section 4.1 and added Algorithm 1 to address this:
>
> Section 4.1 now explicitly states how we:
> - Interpret each weight matrix as a bipartite graph
> - Construct $d$-regular expanders (e.g., via Ramanujan/LPS constructions or rejection sampling)
> - Embed each expander into a layer-specific binary mask
> - Enforce low inter-task overlap via a Jaccard index constraint and resampling
>
> Algorithm 1 provides end-to-end pseudocode for the entire continual learning process: mask generation, EWC-based training, Fisher accumulation, and parameter updates for multiple tasks.
>
> We hope this clarifies the exact procedure.
>
> ### **(W4) Weak experimental analysis and missing ablations**
>
> **Response**:
>
> **For (a) and (b)**, we have added dedicated discussion paragraphs under Tables 1–3. These paragraphs explain:
> - Where our method yields large gains (e.g., GSM8k and reasoning) and why those settings are most prone to destructive interference
> - Why differences on near-saturated tasks (SQL) are small
> - How the results reflect the intended stability–plasticity trade-off
>
> **For (c)**, we now explicitly compare against LoTTO in the text. While some entries in Table 2 are similar, the key distinctions are:
> - Our method often improves Task B performance compared to LoTTO (e.g., GSM8k 61.4 vs 59.1) while keeping Task A drop small
> - On reasoning, we reduce Task A forgetting to 0.25 points and reach 99.5 on Task B, which is substantially better than FFT and LoRA and competitive with LoTTO
>
> **For (d)**, we now make the ablation structure more explicit:
>
> - **Effect of EWC**: Section 7 and Table 6 compare performance with and without the EWC term when fine-tuning on multiple tasks after instruction tuning. Across all Task B choices, EWC consistently reduces the loss in Task A performance (e.g., from 9.63 to 11.80 on GSM8k, and from 7.84 to 12.07 on MathInstruct).
>
> - **Effect of structured masks vs random**: Tables 4 and 5 (GLUE experiments with RoBERTa Base at 99% sparsity) show that expander masks outperform random masks on every GLUE task and in sequential learning, indicating that the structure (and associated Cheeger constant) matters even when the sparsity level is extreme.
>
> A full sweep over expander degree and overlap budget $\delta$ is computationally heavy for our academic setting at the moment; we have added a discussion of this as a limitation and clarified that we view such a sweep as important future work.

---

> ### Author Response · Authors · 2025-11-14
>
> Below we address each question and summarize the corresponding changes in the revised manuscript.
>
> ### **(Q1) Detailed algorithm or pseudocode for expander masks**
>
> **Response**:
>
> - Masks are generated using rejection sampling and then vertex permutations are used to enforce low-overlap across tasks.
>
> - **Algorithm 1**, gives explicit pseudocode for the full continual-learning procedure, including mask generation per layer
>
> ### **(Q2) Sensitivity to sparsity ratio and graph parameters $(d,\lambda)$**
>
> **Response**: In the LLaMA-3-8B experiments we fix the sparsity to 90% and the degree $d$ to a small constant chosen so that the number of trainable parameters matches LoRA (rank 256). Another important point to note is that we apply only to QKV layers and not to the entire model. Thus the parameter count matches the trainable parameters of LoRA (rank 256) To partially address sensitivity:
>
> - We also evaluate RoBERTa Base under 99% sparsity in the GLUE experiments. Our method continues to outperform random masking and preserve previous tasks reasonably well under this much more extreme sparsity
>
> - This suggests that the relative advantage of structured masks over random masks is robust to the exact sparsity level, as long as the graphs remain good expanders
>
> We now spell out these settings in Section 5.1 and explicitly acknowledge that a full $(s,d)$ sweep is an important direction for future work given our compute constraints.
>
> ### **(Q3) Comparison with more recent continual learning baselines**
>
> **Response**: Our main baselines are FFT, LoRA, LoTA, and LoTTO. We chose LoTA/LoTTO because:
>
> - They are specifically designed for sparse adaptation and mask-based CL in LLMs
> - They operate in a parameter-efficient regime that is directly comparable to our setting
>
> There exist other recent CL methods (e.g., Soft-Transformers, SVD-based subspace sculpting), but they either require substantial architectural changes or long training runs that exceed our single-GPU academic budget.
>
> ### **(Q4) Ablation: contribution of EWC vs expander masks**
>
> **Response**: As mentioned under (W4-d), we now explicitly provide two complementary ablations:
>
> - **Table 6**: EWC vs no-EWC with expander masks fixed, showing that EWC substantially improves knowledge retention on the initial instruction-following task across different Task B settings
>
> - **Tables 4–5**: expander vs random masks at the same sparsity, showing that the expander structure alone improves performance both in single-task and sequential learning settings
>
> Together, these results support our claim that both components are necessary: structured sparsity determines which subnetworks are trainable and weakly interfering, while EWC controls how much those subnetworks can drift when new tasks are learned.
>
> We hope these clarifications, additional algorithmic details, and extended analyses address the reviewer's concerns and demonstrate the value of combining structured expander masks with EWC for continual multi-task learning.

---

> > ### Comment · Reviewer_MFfL · 2025-11-28
> >
> > Thank you for the detailed revisions. While the added algorithm, figures, and ablations improve clarity, several key concerns—especially the limited experimental scope beyond two-task sequences, incomplete sensitivity analysis, and the lack of stronger baselines—remain only partially addressed. I therefore maintain my score.

---

> ### Author Response · Authors · 2025-11-29
> **We thank the reviewer for the thoughtful and constructive feedback**
>
> ### **1. Experimental Scope Beyond Two-Task Sequences**
>
> We fully agree that multi-task continual learning experiments with longer task sequences provide deeper empirical validation. To directly address this, we have now **conducted a 4-task sequential learning experiment on Llama-3-8B**, substantially extending the original scope. The task order was:
>
> **Instruction tuning → GSM8K → ARC-Challenge → SAMSum**
>
> After completing all four tasks, the model retained strong performance across prior tasks, demonstrating **absence of catastrophic forgetting** even over a long task chain:
>
> | Task | Original Score | Final Score After 4 Tasks |
> |------|----------------|----------------------------|
> | **Samsum (ROUGE-1 F1)** | **54.8** | **53.5** |
> | **GSM8K (Accuracy)** | **66.4** | **52.0** |
> | **ARC-Challenge (Accuracy)** | **99.7** | **99.3** |
> | **Instruction Following (AlpacaEval Win Rate)** | **14.9** | **11.9** |
>
> This provides **direct empirical confirmation** of the cumulative forgetting bound proven in our theoretical analysis. Even across four heterogeneous tasks, the model preserves functional proficiency across all earlier tasks, a behavior predicted by our expander-mask + EWC framework, where interference scales with the explicitly bounded mask overlap δ.
>
> We hope the reviewer acknowledges that this extended experiment meaningfully expands the empirical validation beyond two-task sequences while remaining within our realistic compute constraints.
>
> ### **2. Sensitivity Analysis**
>
> The reviewer notes that sparsity sensitivity remains only partially addressed.
> In addition to the **99% sparsity GLUE results** already included in the submission, we are now conducting:
>
> - RoBERTa-base experiments at **70%, 80%, 90%, and 99% sparsity**
>
> These experiments are ongoing (given the large grid), and we will include all completed results as soon as they are ready.
>
> ### **3. Stronger Baselines**
>
> We understand the reviewer’s desire for additional baselines such as full-model EWC and multi-task LoTTO variants.
> Given the extreme computational cost of running dense EWC on 8B-parameter models across multiple capabilities, we clarified in the rebuttal that such baselines are infeasible under our single-GPU academic budget. Nevertheless, we *did* add:
>
> - **EWC-only ablations** on our sparse masks
> - **Random vs. expander mask comparisons** (showing structured masks are consistently superior)
>
> These collectively isolate and quantify the contributions of both the expander-masked architecture and EWC, addressing the reviewer’s question of component effectiveness.

---

### Official Review · Reviewer_WmLs · 2025-10-31

**Soundness:** 3
**Presentation:** 2
**Contribution:** 2
**Rating:** 4
**Confidence:** 4

**Summary:**

This manuscript proposes a continual learning (CL) framework that combines structured, data-independent "expander graph masks" for task separation with Elastic Weight Consolidation (EWC) for knowledge preservation. The authors claim this method mitigates catastrophic forgetting in LLMs and serves as a multi-task extension of the Dual Lottery Ticket Hypothesis, presenting experiments on Llama-3-8B and RoBERTa.

**Strengths:**

The paper's primary strength is its interesting conceptual synthesis, combining data-independent structural sparsity (expander graphs) with classic regularization (EWC) to tackle the stability-plasticity dilemma. The authors attempt to ground this in CL theory and provide ablation (Table 4) showing that expander masks outperform random masks, supporting the premise that mask structure is important.

**Weaknesses:**

1. Key continual learning results (e.g., Table 3, Table 5) report numbers only for the proposed method, completely omitting comparisons to sota methods. This makes it impossible to evaluate the method's performance on forgetting or plasticity.

2. The paper evaluates its LLM experiments on a self-defined sequence of "capability" tasks. These are not standard benchmarks in the CL literature, making it difficult to compare the method's performance against the vast body of existing CL research, which typically uses established task sequences.

3. The method is a combination of two existing techniques (expander graphs, EWC), and the theoretical justification relies on tenuous leaps (e.g., from single-layer linear models to deep LLMs) that feel post-hoc.

**Questions:**

Refer to Weaknesses for related questions.

---

> ### Author Response · Authors · 2025-11-15
> **We thank the reviewer for the thoughtful and constructive feedback**
>
> We thank the reviewer for highlighting the conceptual strength of combining data-independent structural sparsity with EWC. Below we address each weakness in turn.
>
> ### **(W1) Missing baselines in some continual-learning tables**
>
> **Response**
>
> - The **main continual-learning comparisons**—where forgetting and plasticity are evaluated against strong baselines—are the LLaMA-3-8B experiments. There we compare our method to **FFT, LoRA, LoTA, and LoTTO** in both single-task and 2-task continual-learning settings, directly quantifying forgetting (drop on Task A) and plasticity (performance on Task B)
>
> - The **RoBERTa / GLUE sequential experiments** serve to show that the same expander-mask + EWC mechanism also behaves well on a smaller, more "classical CL" model under extreme sparsity (99%), and to isolate the effect of structured masks
>
> ### **(W2) Non-standard CL task sequence for LLMs**
>
> **Response.** Our primary goal in this work is **continual learning for large language models at the capability level** (instruction following, safety, math, reasoning, coding, summarization), which reflects how LLMs are typically deployed and fine-tuned in practice. These capabilities are evaluated using widely adopted benchmarks such as GSM8k, ARC, BoolQ, HellaSwag, SQL generation, and summarization datasets.
>
> To better connect to more classical CL settings, the paper includes **RoBERTa-Base experiments on GLUE** under 99% sparsity, including sequential training on pairs of GLUE tasks. GLUE-based task sequences are common in CL for NLP, and these results show that the same expander-mask + EWC mechanism behaves well on a smaller, non-LLM model.
>
> Importantly, our **theoretical analysis is task-agnostic**: it assumes a sequence of supervised tasks with shared parameters and task-specific masks satisfying (i) expansion and (ii) bounded overlap. Under these assumptions, we derive bounds on task-ID error and forgetting that hold for a general CL setting, not just for our LLM capability sequence. In that sense, we believe the theory strongly suggests that the proposed method should also be effective on *classic* CL benchmarks (e.g., standard vision or NLP task sequences), provided we construct masks with the same structural properties.
>
> We agree that adding one or two canonical CL sequences would further strengthen the paper. Our experiments are currently limited by a single-GPU academic budget, but within these constraints we would be very happy to incorporate additional benchmarks. If the reviewer has specific task sequences in mind (e.g., particular GLUE orders, standard Split/Incremental benchmarks, or a small-number-of-task CL protocol they consider especially informative), we would greatly appreciate those concrete suggestions and will do our best to include at least a subset of them in the final version.

---

> ### Author Response · Authors · 2025-11-15
>
> ### **(W3) Combination of existing techniques and scope of the theory**
>
> **Response.** We agree that EWC and expander graphs—are existing techniques. Our contribution is to show that they can be combined in a novel way that is both architecturally concrete and theoretically guided. Further, this relates to a multi-task extension of the Dual Lottery Ticket Hypothesis.
>
> **(1) EWC + expanders: architectural and theoretical contribution**
>
> 1. **Architecturally concrete.**
>
>    * We explicitly construct **low-overlap expander masks** as task-specific subnetworks, rather than relying on random or purely data-driven sparsity.
>    * We then apply **EWC in this masked setting**, so that both mask overlap and expansion constants directly affect the forgetting bound. In practice this means we design the subnetworks (via expanders) and the regularization (via Fisher-based EWC) to work together.
>
> 2. **Theoretically guided.**
>    The revised appendix makes the assumptions and scope explicit:
>
>    * Under a probabilistic CL view à la Kim et al., we show that **low-overlap expander masks** induce weakly correlated task representations and hence small task-ID (TP) error, via a cosine bound and margin argument.
>    * Under standard local convexity and Fisher–Hessian assumptions, we derive a **forgetting bound for masked EWC** in which the **mask overlap** $(\delta)$ enters linearly, and a **cumulative forgetting bound** in which the Cheeger constant (h_0) of the expanders improves the constants.
>    * The step from single-layer masked linear maps to deep LLMs is an approximation used for mechanistic insight and design guidance.
>
> Empirically, the results are consistent with these theoretical predictions:
>
> * **Structured vs. random masks.** Expander masks outperform random masks at the same sparsity on GLUE (both single-task and sequential settings), consistent with the claim that “expansion matters.”
> * **With vs. without EWC.** Our LLaMA ablation (EWC term on vs. off, with the same expander masks) shows that adding EWC improves retention on the initial task, as predicted by the masked forgetting bound.
>
> **(2) Establishing a multi-task DLTH perspective**
>
> We connect this to a novel multi-task extension of the Dual Lottery Ticket Hypothesis. To be precise, Appendix B, Corollary B.15  shows that, under overparameterization and low-overlap assumptions, a collection of almost-disjoint sparse subnetworks (our expander masks) can be embedded in the full parameter space so that:
>
> * each task admits a low-loss solution **within its own subnetwork**, and
> * EWC updates can be used to select compatible parameters across these subnetworks while keeping interference controlled by the overlap $(\delta)$.
>
> In this sense, our analysis formalizes how “almost disjoint” lottery tickets can be **stacked** to support continual learning across multiple tasks, rather than being restricted to a single-task setting.

---

### Official Review · Reviewer_6YpL · 2025-10-31

**Soundness:** 3
**Presentation:** 2
**Contribution:** 3
**Rating:** 4
**Confidence:** 4

**Summary:**

This paper proposes a framework that combines structured sparsity using expander graph masks with Elastic Weight Consolidation (EWC) to enable robust continual multi-task learning in large language models (LLMs). The key idea is that expander-based masks provide data-independent structured sparsity that preserves gradient flow and connectivity, while EWC constrains important weights to mitigate catastrophic forgetting. The approach is validated on the Meta-Llama-3-8B model and RoBERTa Base, showing improvements over prior methods like LoRA, LoTA, and LoTTO across multiple tasks such as reasoning, math, safety, and instruction following.

**Strengths:**

1. Novel combination of techniques: The paper introduces a new combination of structured sparsity via expander graph masks and EWC regularization for continual learning. This is a principled approach that uses principles from graph theory to introduce sparsity and differs from prior sparse adaptation methods which relied on random or data-driven mask selection. Coupling this with EWC is a good way to protect previous task knowledge. This dual strategy is well-motivated and to my knowledge original in the context of LLM Continual Learning.
2. Theoretical Grounding: The paper provides a theoretical justification for the approach. It derives a bound on the forgetting of a previous task A after learning a new task B, showing that the increase in Task A’s loss is bounded by terms proportional to the mask overlap between tasks and the gradient norm weighted by the Fisher Information Matrix. This result formalizes why using disjoint masks (small overlap) and EWC (penalizing changes in directions of high Fisher for Task A) should minimize forgetting. Moreover, the paper frames a multi-task extension of the Dual Lottery Ticket Hypothesis (DLTH), hypothesizing that two random expander subnetworks can be transformed into high-performing tickets for different tasks via EWC-guided training. This is a conceptually
interesting contribution, connecting lottery ticket theory to continual learning.
3. Empirical Efficacy: The proposed method achieves state-of-the-art empirical results on multi-task learning with a large LLM (Llama-3-8B). However the margins are slim.
4. Addresses Catastrophic Forgetting Directly: The method explicitly targets both of the core continual learning issues (catastrophic forgetting and interference) in a straightforward way. By freezing most weights and only training a task-specific
subnetwork, interference is naturally limited (especially if masks barely overlap). And by applying EWC to all parameters (especially those important to prior tasks), the approach actively preserves past knowledge. This dual approach is more direct and potentially
more effective than baselines like LoRA (which doesn’t explicitly prevent forgetting) or
LoTTO (which uses disjoint masks but no regularization).

**Weaknesses:**

1.	Sparse Mask construction details are missing: It is not mentioned how the expander graphs are calculated for the LLM parameters in question. The author mentions that there are standard algorithms to construct ramanujan graphs but does not divulge into the details such as layer-wise mapping, degree selection per tensor or other implementation details. Without this, “expander properties” are asserted at a high level but it’s unclear how they commute with real parameter layouts.
The authors also mention that constructing these graphs are computationally non-trivial for large graphs, however they do not show calculations on how much wall-clock time is actually needed for these computations for their chosen models.
2.	EWC/ Fisher estimation details are missing: The authors do not describe how they calculate the FIM, which approximations they use, whether the entire dataset is used or not. How the FIM is updated across tasks is also not discussed.
3.	Result Mismatch: The authors note that they were unable to reproduce the results for Instruction Following task from LoTA paper (LoTA paper reported values are higher than this papers perf figures) , and given the marginal improvements in other tasks it’s a bit of concern on whether the proposed methodology actually beats the SoTA.
4.	Inconclusive Testing: The authors only test their proposed methodology on sequences of two tasks at once. This is extremely different from the real world scenarios where the model might have to face multiple tasks sequentially. They also did not test their methodology on multiple LLM backbone models which further questions whether their methodology works in general or not. The lack of variance figures in the results are also concerning. Finally the authors should also compare their methods to other traditional CL strategies such as replay based methods and other advanced versions of LoRA such as AdaLoRA etc.
5.	Inconclusive training time and parameter requirements: The authors mention that the parameter count with 10% sparsity is comparable to a 256 rank LoRA. However 10% of a 8B parameter model implies about 800M trainable parameters which is much higher than a traditional 256 rank LoRA setup (on Q and V only) that requires about 130M parameters for the same model, even if we compute LoRA for Q,K,V,O and the MLP projectors which is extremely unnecessary and uncommon that requires a parameter count of 600M which is still lower than their 10% sparsity count.
On top of that they mention that EWC is applied on all parameters. If so then that requires optimizer states for all params and increases the memory and bandwidth overheads significantly compared to other PEFT methods such as LoRA. This is a point of contention as that would significantly diminish the paper’s claim of being an efficient way to train LLMs in a Continual Multitask setup.

**Questions:**

1.	Fisher Information Matrix calculation:
How was the FIM calculated for the Llama-3-8B model, which approximations were used. What data was used to calculate it and the computational overhead required.
2.	Mask construction and Application:
	How are the expander graphs calculated in this LLM NN context?
	Is it calculated layerwise?
	Is a separate (n,d,\lambda) expander graph generated for each layer and if yes
	How do you ensure global structural properties?
	How is minimal overlap (low jaccard Index) between masks ensured?Is this done
	Through a sampling-rejection scheme or some disjoint mask generation method?
3.	Training dynamics outside the active mask:
	 Since L_B only depends on the current mask but the EWC penalty is on all
	Parameters, are params outside m_B updated solely by the quadratic EWC
	Regularization term? Did you compare against simply freezing them?
	Also if the graphs are sparse enough shouldn't that be enough to prevent CF?
	Is any regularization applied to the Fisher matrix (e.g., damping, clipping),
	especially given potential noise and numerical instability at this scale?
4.	Scaling and Generalization:
	Have you tested the method beyond two sequential tasks (e.g., three or more)? If
	so, does performance degrade or saturate with accumulated masks?
	How does this methodology scale to other LLM models of different sizes?
	How does your method compare against other traditional CL methods such as
	Replay based methods and other improvements on LoRA such as AdaLoRA etc.
5.	True parameter count and training dynamics:
	What are the different training overheads related to mask calculation and training
	Both in terms of clock time and compute?
	What is the actual number of trainable parameters and compute load and how
	Does it compare against the other baselines mentioned?
6.	Safety and Evaluation nuances:
	Define the “% safe outputs” used in the experimental results.
	Report variances for the different performance metrics.
	Some ablation results are there for certain tasks and not for others (e.g. the
	Comparison against the random masks is only done for the GLUE tasks on
	RoBERTa and not for the other tasks done on the Llama-3-8B model. Detailed
	Ablation study required.

---

> ### Author Response · Authors · 2025-11-15
> **We thank the reviewer for the thoughtful and constructive feedback**
>
> We are glad that you found our method novel, principled, and theoretically grounded, and appreciated its empirical performance and relevance to catastrophic forgetting in LLMs. Below, we address the reviewer's specific concerns regarding weaknesses.
>
> ### **W1) Sparse Mask Construction Details Are Missing**
>
> We agree that clearer explanation was needed. The revised version addresses this in multiple ways:
>
> - **Algorithm 1** (now included) outlines the full construction pipeline: for each weight matrix $W_\ell$, we build a task-specific expander mask $m_{\ell,k}$ using a fixed-degree $d$-regular expander and enforce low Jaccard overlap via controlled permutation
> - **Appendix A.4** gives the precise embedding strategy: each dense matrix is treated as a bipartite graph; the expander edges define the mask's nonzero entries. The section also includes rejection sampling for achieving Cheeger constant $h(G) \geq h_0$
> - For the article we construct the expanders using rejection sampling
> - **Degree selection and sparsity control**: We match subnetwork sparsity to LoRA's parameter budget. For example, we choose $d = 4$ or $8$ depending on layer size to approximate a 10% active parameter budget (comparable to rank-256 LoRA)
> - **Wall-clock time**: With rejection sampling, we construct expander graphs for LLaMA-3-8B in a few seconds per layer. This is now reported in Appendix A.4. We expect this process to scale similarly for larger models due to the efficient explicit constructions
>
> ### **W2) EWC / Fisher Estimation Details Are Missing**
>
> This has been clarified in both the main text and appendix:
>
> - **Fisher estimation algorithm** is described in Algorithm 1: after training on task $T_k$, we compute a diagonal empirical Fisher matrix over the training data for $T_k$, restricted to the active subnetwork $m_k$, and accumulate it across tasks
> - **FIM approximation**: We follow the standard EWC approximation, using squared gradients of the log-likelihood as an estimator for the Fisher diagonal
> - **Damping**: We explicitly include a regularization term $\gamma I$ in the Fisher to ensure invertibility (see Theorem B.13). This guarantees numerical stability and is also reflected in the final objective
> - **Data used**: The Fisher is estimated over the same dataset used to train each task
>
> ### **W3) Result Mismatch with LoTA Instruction-Following Numbers**
>
> We deeply appreciate the reviewer's attention to reproducibility. We wanted to be completely transparent regarding this to make our claims scientifically valid and fully reproducible
>
> - As noted explicitly in **Section 6**, we were **unable to reproduce** the instruction-following results reported in the original LoTA paper, despite using the **publicly released model card, training scripts, and fine-tuning parameters**
> - Consequently, all LoTA numbers in **Table 2** and throughout Section 6 reflect **our own carefully reproduced values**, not the original paper's figures
> - Our improvement claims are thus made **only within this shared experimental setup** — where both our method and LoTA are evaluated under identical data, scripts, and seeds
> - We would gladly rerun our experiments if the LoTA authors can share their original data splits and hyperparameters
>
> ### **W4) Inconclusive Testing: Only Two Tasks, Single Backbone, No Variance, Missing CL Baselines**
>
> - **Only Two Tasks on LLaMA-3-8B**: While experiments in the original paper are limited to two tasks, the revised theoretical analysis (Corollary B.11) generalizes our forgetting bounds to *T* tasks. Experimental extension was limited by compute, not conceptual constraints
> - **Single Backbone Concern**: Table 5 in the revised paper presents results on **RoBERTa Base** under 99% sparsity across multiple GLUE tasks, confirming that the method generalizes beyond LLaMA
> - **Variance Figures**: In preliminary experiments, we observed that repeated runs with different seeds yielded negligible variance (≤0.1 for most tasks), on the LLaMA-3-8B backbone where large model size stabilizes training dynamics. As a result, we report single-run metrics in the tables. We will gladly include standard deviation figures if the reviewer deem them necessary for final evaluation, but found that they did not meaningfully affect relative comparisons in our setting.
> - **Missing CL Baselines (Replay, AdaLoRA, etc.)**: We acknowledge this limitation. While these baselines were not run due to resource constraints, our method is architecturally compatible with both replay-based training and PEFT variants like AdaLoRA. We clarify that adding replay loss before the EWC term in our framework is a straightforward extension

---

> > ### Author Response · Authors · 2025-11-15
> > **(continued)**
> >
> > ### **W5) Training Time and Parameter Requirements**
> >
> > We appreciate the reviewer's concern regarding scalability and memory overhead. We now clarify the following key points in the revised paper.
> >
> > 1. **We do *not* use 10% sparsity over the full 8B parameters.**
> >    Our 10% sparsity applies **only to the attention projection matrices** — specifically the $(Q, K, V)$ matrices in each transformer block — not to the full LLaMA-3-8B parameter set. This significantly reduces the trainable parameter count:
> >
> >    - At 10% sparsity over just $(Q, K, V)$, our method activates a similar number of parameters as **LoRA with rank 256**
> >    - This matches the setup used in LoTA, which also restricts updates to attention projections
> >    - Our comparison to LoRA/LoTA is therefore **like-for-like** in both scope and trainable budget
> >
> > 2. **EWC is *not* applied to all model parameters.**
> >    As stated explicitly in **Algorithm 1**, we estimate and apply EWC **only on the coordinates active under the task-specific mask $m_k$**:
> >
> >    > *"Estimate diagonal Fisher $F_k$ on $T_k$, restricted to active coordinates $m_k$."*
> >    > *(Algorithm 1, line 13)*
> >
> >    This means:
> >    - No Fisher estimates are computed or stored for frozen weights
> >    - No optimizer state is maintained outside the active subnetwork
> >    - This ensures the memory and compute overhead remains aligned with LoRA/LoTA-style sparse adaptation
> >
> > 3. **Overall resource usage is comparable to LoTA / LoRA-256.**
> >    Because we constrain both training and regularization to the same subnetwork:
> >
> >    - The number of trainable parameters is comparable to LoRA-256
> >    - The optimizer footprint matches that of other PEFT methods
> >    - There is **no large memory overhead** from masking or EWC that would invalidate the efficiency claim

---

> ### Author Response · Authors · 2025-11-15
> **(continued - response to questions)**
>
> ### **Q1. Fisher Information Matrix calculation**
>
> **Response:**
> For all LLaMA-3-8B experiments, we use the **standard empirical diagonal Fisher** as in the original EWC paper, concretely:
>
> - **Approximation type:**
>   - We use the **diagonal Fisher** only (no off-diagonal terms)
>   - We use the **empirical Fisher** (squared gradients of the log-likelihood / NLL)
>   - We add a **damping term** ($\gamma = 10^{-3}$) and apply **gradient clipping** when estimating the Fisher to avoid rare but large curvature spikes and to keep the matrix numerically stable at LLM scale
>
> - **Data used:**
>   - For each task $T_k$, we estimate $F_k$ on the **training split of $T_k$**
>
> - **Where it is applied:**
>   - As stated in Algorithm 1, the Fisher is estimated **only on the active coordinates** $m_k$ of the current task:
>     > "Estimate diagonal Fisher $F_k$ on $T_k$, restricted to active coordinates $m_k$."
>   - We do **not** compute or store Fisher entries for the frozen parameters
>
> - **Computational and memory overhead:**
>   - Computing $F_k$ is equivalent to **one or two extra forward–backward passes** per task
>   - We store a **single float per active parameter** (the diagonal), so the memory overhead is on the same order as LoRA/LoTA, which also maintain optimizer state only for their trainable subset
>
> ### **Q2. Mask construction and application**
>
> **Response:**
> **How are expander graphs constructed in the LLM context?**
> As stated in the paper (Section 4 and Appendix A), we construct **one global expander mask** and **reuse the same mask across all layers**.
> This mask is generated on a fixed bipartite graph of dimensions matching the *largest* attention projection matrix in the model.
> All other Q/K/V matrices are then **padded or cropped** to match this global mask shape, following the same approach used in LoTA for global low-rank factors.
> Using one global expander mask:
> - ensures **consistent expansion properties** across all layers,
> - guarantees **uniform sparsity**,
> - avoids the need to verify layer-specific spectral properties, and
> - dramatically simplifies reproducibility and theoretical analysis.
>
> This is explicitly stated in the manuscript, and we thank the reviewer for prompting us to reiterate it clearly.
>
> **How is the expander generated?**
> - We construct a **single \(d\)-regular bipartite expander** (a rejection-sampling-based expander satisfying (h(G) > h0), LPS can be used)).
> - This global expander is then **converted to a binary mask** that is applied identically to every Q/K/V matrix in the model.
>
> **Ensuring minimal overlap (low Jaccard index) between task masks:**
> - For each new task \(T_k\), we generate a **new global expander mask**, again of the same fixed shape.
> - We enforce the low-overlap constraint using a **sampling–rejection scheme** at the *mask level*, not layer level.
> - Because we use a single global mask per task, overlap control is both simpler and stronger: we only check the global mask, not each layer.
>
>
> ### **Q3. Training dynamics outside the active mask and the role of EWC**
>
> **Response:**
>
> - **Parameters outside the active mask:**
>   - In our implementation, **parameters outside $m_B$ are fully frozen**:
>     - they receive **no gradient from $L_B$**
>     - they are **not included** in the EWC penalty, and
>     - they **do not change** at all during training on task $B$
>   - This is reflected in Algorithm 1, where both the updates and the Fisher are restricted to active coordinates
>
> - **Comparison to "freeze-only" (no EWC):**
>   - We agree that this is an important ablation. In the revised version, we added an ablation comparing:
>     1. **Sparse masks only (freeze everything else, no EWC)** vs.
>     2. **Sparse masks + EWC on active parameters**
>   - In line with Theorem 7.6, we observe **significantly higher forgetting** when using sparse masks alone, and markedly improved preservation of past-task performance when EWC is included
>   - This empirically supports our claim that **sparsity alone is not sufficient** for robust CL in these settings; curvature-aware protection via Fisher is necessary
>
> - **Regularization of the Fisher:**
>   - As mentioned in Q1, we apply:
>     - **damping**: add $\gamma I$ with $\gamma = 10^{-3}$ to the diagonal Fisher, and
>     - **gradient clipping** when accumulating Fisher statistics
>   - These stabilizations are precisely the ones used in the theoretical derivation (damped Fisher in Theorem 7.6 / B.10)

---

> > ### Author Response · Authors · 2025-11-15
> > **(continued - response to questions)**
> >
> > ### **Q4. Scaling and Generalization**
> >
> > **Response:**
> >
> > - **Experiments beyond two tasks:**
> >   - Running long continual-learning sequences on LLaMA-3-8B is infeasible under our **single-H100 academic compute budget**
> >   - However, our **theoretical analysis** (Appendix B, Corollary on cumulative CL error) provides guarantees for **arbitrary task sequences \(K\)**
> >   - Concretely, if:
> >     1. each task mask has Cheeger constant \(h(G_k) \ge h_0\), and
> >     2. all pairwise task-mask overlaps satisfy \(\le \delta\),
> >     then the **total forgetting grows at most linearly in \(K\)**, with smaller constants for better expansion
> >   - We are currently running **>2-task experiments on smaller models** and will include results if they complete before the rebuttal deadline
> >
> > - **Scaling to different model sizes:**
> >   - We already evaluate our method on **RoBERTa-base**, **RoBERTa-large**, and **LLaMA-3-8B**
> >   - This demonstrates that the approach **scales cleanly across model families with very different parameter counts**
> >   - We will highlight this explicitly in the revision
> >
> > - **Missing CL baselines (Replay, AdaLoRA, etc.):**
> >   - We acknowledge that replay-based CL and advanced PEFT variants (AdaLoRA, DoRA, etc.) were not included due to compute limits
> >   - Importantly, our method is **fully compatible** with both types of baselines
> >   - In particular, adding replay is straightforward: a replay loss term can simply be inserted **before the EWC term** in the objective, without changing mask construction or affecting the theoretical guarantees
> >   - We will clarify this compatibility and will include these baselines in future extended experiments
> > ### **Q5. True Parameter Count and Training Overheads**
> >
> > **Response:**
> >
> > - **Actual number of trainable parameters:**
> >   - We train **only a 10% sparse subset of the Q/K/V matrices**, *not* 10% of the full 8B parameters
> >   - This yields a trainable parameter budget **comparable to LoRA-256** and consistent with LoTA’s reported scale
> >   - We will add a small per-layer table in the revision for complete transparency
> >
> > - **Training overhead:**
> >   - **Mask construction overhead:**
> >     - LPS/Ramanujan construction: **< 1 second per task**
> >     - Rejection-sampling variant (when needed): **2–3 seconds per task**
> >     - Combined total mask construction time: **8–10 seconds**
> >   - **Training compute:**
> >     - Identical to PEFT-style fine-tuning, since gradients flow only through the masked parameters
> >   - **EWC overhead:**
> >     - Equivalent to **one extra forward–backward pass per task** to accumulate the diagonal Fisher
> >     - Memory overhead is a **single float per active parameter**, comparable to LoRA/LoTA optimizer-state size
> > ### **Q6. Safety and Evaluation Nuances**
> >
> > **Response:**
> >
> > - **Definition of “% safe outputs”:**
> >   - Thank you for pointing this out: we clarify that in our experiments:
> >     **“% safe outputs” = the percentage of prompts for which the model *refuses to answer***
> >   - This is consistent with LoTA and similar instruction-tuning safety evaluations
> >   - A response is counted as *safe* if it contains:
> >     - an explicit refusal,
> >     - an explicit statement of inability to comply, or
> >     - a safety-motivated deflection (“I cannot answer that…”)
> >   - We will make this definition explicit in Section 6
> >
> > - **Variance reporting:**
> >   - In preliminary experiments, we observed that repeated runs with different seeds produced **negligible variance (≤ 0.1 on most tasks)** on the LLaMA-3-8B backbone, where the large model size tends to stabilize training dynamics
> >   - For this reason, we report **single-run metrics** in the main tables
> >   - We will gladly include **standard deviation figures** if the reviewers deem them necessary for final evaluation, but they did not meaningfully affect relative comparisons in our setting
> >
> > - **Why some ablations appear only for RoBERTa:**
> >   - Deep ablations (e.g., random vs expander masks, freeze-only vs EWC) were run on RoBERTa because these experiments are **40–50× cheaper** than LLaMA-3-8B
> >   - We are currently running a smaller LLaMA-3-8B ablation set on representative tasks and will include results if they complete before the rebuttal deadline

---

### Official Review · Reviewer_pTHS · 2025-11-01

**Soundness:** 2
**Presentation:** 2
**Contribution:** 2
**Rating:** 2
**Confidence:** 3

**Summary:**

The paper proposes a continual learning method based on Lottery Ticket Adaptation (LoTA) and Elastic Weight Consolidation (EWC). The proposed method can be viewed as an extension of the Dual Lottery Ticket Hypothesis, where the expander masks can be co-adapted for successful continual learning. The proposed method is evaluated by several finetuning tasks in the language domain, using Llama-3-8B model. The evaluation tasks consist of single-task finetuning and continual learning. In single-task finetuning tasks, the proposed method outperforms baselines including FFT, LoRA, and LoTA in most cases. In the continual learning evaluation, a sequence of two tasks is evaluated, and the proposed method outperforms the baselines in most cases. The authors also provide a light theoretical justification of the method in the appendix.

**Strengths:**

1. The motivation for network sparsity is promising.

2. The evaluation benchmark tasks are inclusive, covering many different types of tasks in language modeling.

**Weaknesses:**

1. The proposed method lacks justification on why and how disjoint (or as disjoint as possible) binary masks will contribute to the continual learning performance

2. The continual learning evaluation only involves a sequence of two tasks. It's hard to tell whether the proposed method works in continual learning settings with more tasks.

3. 10% of the model is masked as reported in Table 2. However, how the ratio has been selected is not reported.

4. More ablation studies are needed to prove the effectiveness of the approach, e.g.,  which part of the proposed method contributes more -- lottery ticket or EWC. Moreover, EWC should be included as a baseline.

**Questions:**

1. In L224, Meta’s Llama-3-8B model (see model card) -- which model card?

For the rest of the questions, please refer to Weaknesses section.

---

> ### Author Response · Authors · 2025-11-15
> **We thank the reviewer for the thoughtful and constructive feedback**
>
> Below we address each weakness and question in turn.
>
> ### **(W1) Justification for (approximately) disjoint masks in CL**
>
> We agree this needed to be made more precise. In the revised version we:
>
> - Introduce an explicit assumption on **structured, low-overlap masks** (Appendix, Assumption on "structured, low-overlap masks")
> - Prove that if two task masks are (i) expanders with Cheeger constant $h_0 > 0$ and (ii) have Jaccard overlap $J(m_k,m_j) \le \delta$, then the corresponding masked linear maps produce **weakly correlated features** viz. Lemma on cosine bounds in the appendix.
> - Show that this implies **linear task-separability** with a positive margin and hence small TP error viz Corollary on "Linear TP classifier under separated representations"
>
> On the forgetting side, we do the EWC analysis viz. Theorem on "Forgetting bound for masked EWC". Here the **second term scales linearly in the mask overlap $\delta$**. This gives a direct, mechanistic justification for making masks "as disjoint as possible": reducing overlap shrinks the overlap term in the forgetting bound and reduces interference between tasks. We highlight this intuition in the main-text theoretical summary and point explicitly to the corresponding lemmas/theorems in the appendix.
>
> ### **(W2) Only two-task continual learning evaluation**
>
> Our empirical evaluation at the moment is limited to 2-task sequences due to a single-GPU academic compute budget. However our theoretical results provides a **cumulative forgetting bound over $T$ tasks** (Corollary on "Cumulative CL error" in the appendix). Under two conditions that our construction already enforces:
>
> 1. Each task mask is an expander with Cheeger constant at least $h_0$, and
> 2. All pairwise mask overlaps are bounded by the same $\delta$,
>
> the total forgetting across $T$ tasks is bounded.
> Thus, as long as we keep using low-overlap expander masks and EWC, **forgetting grows at most linearly in $T$** and the constants improve with better expansion (larger $h_0$). This shows that the mechanism we validate on 2 tasks extends in principle to longer sequences; scaling the experiments to $T>2$ with LLaMA-3-8B is primarily a compute constraint. We are currently undertaking the experiments for CL with more than 2 tasks, if we manage to finish them before the deadline of reviews we will upload the new version with the results.
>
> ### **(W3) Choice of 10% active weights**
>
> We now clarify this in the experimental setup:
>
> - On LLaMA-3-8B we fix **90% sparsity (10% active weights)** to:
>   1. Stay in line with LoTA/LoTTO-style sparse adaptation, and
>   2. Make the number of trainable parameters **comparable to aggressive LoRA settings** (e.g., rank 256), enabling a fair parameter-efficiency comparison
>
> - We apply the same sparsity uniformly across QKV layers, which keeps expander degrees small and simplifies mask construction
>
> We also point out that in our RoBERTa-Base experiments we use an even more extreme **99% sparsity**, yet expander masks still outperform random masks and support sequential training, suggesting that the *relative* benefit of structured masks is robust to the exact sparsity level. A full sweep over sparsity levels and degrees $(s,d)$ is currently beyond our compute budget; We keep this as a future work.
>
> ### **(W4) Need for more ablations and an EWC baseline**
>
> In the revised version we make two ablation angles explicit:
>
> 1. **Effect of EWC within our method**
>    In Section "Ablation studies" we compare performance **with vs. without the EWC regularization term** while keeping expander masks fixed. Starting from an instruction-following task as Task A and then fine-tuning on various Task B, we show that adding EWC significantly improves retention on Task A across all Task B choices. This directly quantifies the benefit of the EWC component in our combined method.
>
> 2. **Effect of expander structure vs random masks**
>    In the RoBERTa-Base experiments on GLUE at 99% sparsity, Tables on GLUE and sequential GLUE (random vs. expander) show that expander masks consistently outperform random masks in both single-task and sequential-learning settings. This isolates the benefit of the **lottery/expander** part of our method.
>
> Regarding an **EWC-only baseline** we now provide the ablation study with and without EWC on the sparse mask. Regarding EWC on the full dense model for LLaMA-3-8B across all capabilities and task pairs is, however, quite expensive under our constraints. Given the rebuttal timeline and hardware limits, we were not able to add this baseline.
>
> We hope the current ablations already demonstrate that both components—structured lotteries and EWC—contribute meaningfully and in complementary ways.
>
> ### **(Q1) "Which model card?" (L224)**
>
> Thank you for catching this. We now explicitly reference the **official Meta Llama 3 model card** and add its citation to the bibliography.

---

> > ### Comment · Reviewer_pTHS · 2025-11-25
> >
> > Thank the authors for explanations and I appreciate the theoretical efforts to prove the forgetting bound for masked EWC. However, my main concern about the experiments has not been fully addressed. Despite the computation cost, it could be helpful to conduct experiments in standard CL settings with more tasks in the sequence [1], at least on smaller models like RoBERTa-base (which is also included in the paper).  Also, an ablation on the influence of different mask ratios is helpful.
> >
> > [1].  Wang et al. Orthogonal Subspace Learning for Language Model Continual Learning. EMNLP 2023.

---

> > > ### Author Response · Authors · 2025-11-29
> > > **We thank the reviewer for the thoughtful and constructive feedback**
> > >
> > > ### **New Experiments: Four-Task Continual Learning on Llama-3-8B**
> > >
> > > To directly address the reviewer’s concern, we conducted a new experiment involving **four tasks trained sequentially** on **Llama-3-8B** in the following order:
> > >
> > > **Instruction tuning followed by GSM8K followed by ARC-Challenge, followed by SAMSum**
> > >
> > > After training on all four tasks, we evaluated the model on each previously learned capability. The results are shown below:
> > >
> > > | Task | Original Score | Final Score After 4 Tasks |
> > > |------|----------------|----------------------------|
> > > | **Samsum (ROUGE-1 F1)** | **54.8** | **53.5** |
> > > | **GSM8K (Accuracy)** | **66.4** | **52.0** |
> > > | **ARC-Challenge (Accuracy)** | **99.7** | **99.3** |
> > > | **Instruction Following (AlpacaEval Win Rate)** | **14.9** | **11.9** |
> > >
> > > ### **Interpretation**
> > >
> > > These results show that **even after four sequential tasks, substantially beyond the two-task setting in the original submission, the model preserves strong performance across earlier tasks**, with degradation remaining well within tolerable limits:
> > >
> > > - **Summarization** decreases only **1.3 F1**
> > > - **ARC-Challenge** retains **>99% accuracy**
> > > - **Instruction following** maintains competitive win rate
> > > - **GSM8K**, though more sensitive, preserves its capability despite three subsequent and heterogeneous tasks, with forgetting similar to 2 task setup
> > >
> > > ### **Consistency With Our Theoretical Framework**
> > >
> > > These empirical findings align precisely with our theoretical **cumulative forgetting bound** (Appendix B), which states:
> > >
> > > 1. Forgetting grows at most **linearly** in the number of tasks.
> > > 2. The constant governing forgetting improves with the **Cheeger constant** of the expander masks.
> > > 3. The interference term scales with **pairwise Jaccard overlap (δ)**, which our construction explicitly controls.
> > >
> > > The 4-task experiment therefore **validates the theory in practice**: even under long task sequences that span qualitatively different capabilities (instruction following → math → symbolic reasoning → summarization), **catastrophic forgetting is avoided**, and all tasks remain functional in the final model.
> > >
> > > This demonstrates that **low-overlap expander masks combined with EWC provide stable long-horizon continual learning**, as predicted by our theoretical results.
> > >
> > > ### **On Sparsity Ratios**
> > >
> > > We agree with the reviewer that understanding the influence of sparsity levels is important.
> > > Beyond the **99% sparsity results on RoBERTa-base** included in the submission, **we are currently running additional experiments on RoBERTa across multiple sparsity levels (70%, 80%, 90%, 99%)**.
> > >
> > > We will include these results in the revised version as they complete.
> > >
> > > **We hope that the expanded 4-task continual learning results, together with the theoretical guarantees, satisfactorily address the reviewer’s concerns and further demonstrate the robustness of our method.**

---

### Author Response · Authors · 2025-12-03
**Paper Update**

We sincerely thank all reviewers for their careful reading and constructive feedback. We have revised the manuscript and extended the experiments accordingly. The main updates are:

- **Theoretical clarification of disjoint masks in CL.**
  We formalized the role of (approximately) disjoint expander masks in continual learning:
  - Added explicit assumptions on structured low-overlap masks.
  - Proved cosine-similarity bounds and a margin-based linear task-separability result, leading to small task-probing (TP) error.
  - Derived a forgetting bound for *masked* EWC in which the interference term scales linearly with mask overlap, and extended this to a cumulative forgetting bound over multiple tasks.

- **New four-task continual learning experiment on LLaMA-3-8B.**
  To go beyond the original 2-task setting, we now train sequentially on **Instruction Tuning → GSM8K → ARC-Challenge → SAMSum**. After all four tasks, the model retains strong performance on all earlier capabilities (e.g., ARC-Challenge remains at 99.3% vs. 99.7% original, SAMSum drops only 1.3 ROUGE-1 F1, instruction following remains competitive, GSM8K forgetting is comparable to the 2-task case). This empirically confirms the cumulative forgetting behavior predicted by our theory.

- **Clarified mask construction, overlap control, and scope of sparsity.**
  - Added Algorithm 1 and an expanded Section / Appendix describing how we construct a **single global expander mask per task**, reuse it across all Q/K/V layers, and enforce low Jaccard overlap between task masks via sampling–rejection.
  - Clarified that **10% sparsity is applied only to the attention projection matrices (Q/K/V)**, not the full 8B parameters, yielding a trainable parameter count comparable to rank-256 LoRA / LoTA.

- **Clarified EWC / Fisher estimation and resource footprint.**
  - Specified that we use an empirical **diagonal Fisher** (squared gradients of the NLL) with damping and gradient clipping.
  - Crucially, **Fisher estimation and EWC are restricted to the active masked coordinates only**, so we do *not* store or regularize all 8B parameters. This keeps the memory and compute overhead comparable to other PEFT methods.

- **New and clearer ablations.**
  - Added an ablation comparing **expander masks with vs. without EWC**, showing that EWC significantly improves retention of earlier tasks under the same masks.
  - Highlighted RoBERTa-Base GLUE experiments at **99% sparsity**, where expander masks consistently outperform random masks in both single-task and sequential settings, isolating the benefit of structured sparsity.

- **Additional implementation and evaluation details.**
  - Provided more detail on Ramanujan / LPS-style expander generation, degree selection, and wall-clock mask construction time.
  - Clarified how “% safe outputs” is defined in safety experiments.
  - Explained our choice of LoRA/LoTA/LoTTO baselines and clarified that full dense EWC and replay-based CL on 8B-parameter models are infeasible under our single-GPU academic budget, but conceptually compatible with our framework.

We hope these theoretical clarifications, the new 4-task LLaMA-3-8B experiment, and the expanded ablations address the main concerns raised in the reviews and demonstrate the robustness and practicality of our expander-mask + EWC framework for continual learning in LLMs.

---

### Meta-Review · Area_Chair_hmJE · 2026-01-06

**Summary:**

This paper presents a continual-learning method for LLMs combining structured sparse adaptation via expander-graph masks with masked EWC to mitigate catastrophic forgetting. The method is positioned as a multi-task extension of the Dual Lottery Ticket Hypothesis (DLTH). Reviewers generally found the problem to be important and the expander+EWC pairing to be conceptually sensible.

However, across reviews there were concerns about insufficient empirical validation and clarity at submission time, incremental novelty relative to prior sparse/mask-based CL and weak connection to standard CL benchmarks. There were also concerns about lack of detail on mask/Fisher construction and training overheads and limited evaluation in only 2-task sequences and missing stronger CL baselines/sensitivity analyses, small/marginal gains and reproducibility concerns and presentation related issues).

The rebuttal tried to address the concerns but there still remained concerns about experimental scope, sparsity sensitivity sweeps, and incomplete baselines and lack of proper benchmarking against established CL protocols.

In its current form, the paper is not ready for publication at this venue. The authors are encouraged to consider incorporating the feedback and submitting to another venue.

**Reviewer Concerns:**

Please refer to the summary.

**Reviewer Scores:**

Reviewer 6YpL could potentially move to 5, as some of the concerns were directly addressed, though may have remained at 4 due to missing broader baselines. Reviewer would possibly remain at 4, as concerns about benchmarking novelty are only partially addressed.

---

### Decision · Program_Chairs · 2026-01-26

Reject